# INTER-ENVIRONMENTAL WORLD MODELING FOR CONTINUOUS AND COMPOSITIONAL DYNAMICS

## ABSTRACT

Various world model frameworks are being developed today based on autoregressive frameworks that rely on discrete representations of actions and observations, and these frameworks are succeeding in constructing interactive generative models for the target environment of interest. Meanwhile, humans demonstrate remarkable generalization abilities to combine experiences in multiple environments to mentally simulate and learn to control agents in diverse environments. Inspired by this human capability, we introduce *World modeling through Lie Action* (WLA), an unsupervised framework that learns continuous latent action representations to simulate across environments. WLA learns a control interface with high controllability and predictive ability by simultaneously modeling the dynamics of multiple environments using Lie group theory and object-centric autoencoder. On benchmark synthetic and real-world datasets, we demonstrate that WLA can be trained using only video frames and, with minimal or no action labels, can quickly adapt to new environments with novel action sets.

## 1 INTRODUCTION

Originally proposed as a framework to support automatic planning and decision-making, world models (Ha & Schmidhuber, 2018) predict future states conditioned on actions, allowing agents to anticipate the consequences of their interactions. There have been extensive efforts in the development of world models and related learning methods. Many of them rely on *discrete* representations of actions and/or images (Hafner et al., 2023; Hu et al., 2023) in order to leverage the techniques of autoregressive inference that are extensively developed in language modeling. Recent works such as Genie (Bruce et al., 2024) and LAPO (Schmidt & Jiang, 2023) have advanced the field by learning interactive world models with few/no explicit action labels. These models have shown promising results in capturing complex dynamics and discrete actions in specific environments.

Humans, however, possess an extraordinary ability to generalize learned skills across diverse environments by leveraging *continuous* and *compositional* representations of actions. For example, after mastering basic movements in a few 2D action-adventure games, a person can quickly adapt to a game of a different type(e.g. Pac-Man) by leveraging the knowledge of common 2D concepts such as moving in continuous directions (Schmidt, 1975; Poggio & Bizzi, 2004). Continuous action representations allow for smooth transitions and fine-grained control, while compositionality enables complex actions to be built from simpler primitives, facilitating generalization and transfer learning (Flash & Hochner, 2005; Botvinick & Plaut, 2004).

Inspired by this human capability, we hypothesize that, in order to learn an interactive world model that generalizes across environments, it is essential to construct an environment-agnostic simulator that embraces continuous and compositional action representations. In this paper, we introduce **World modeling through Lie Action (WLA)**, an unsupervised framework that aims to learn such a simulator. Our approach models the nonlinear and continuous transition operators in the observation space with a Lie group that acts linearly on partitioned latent vector spaces, where each partition corresponds to different objects and fundamental action axes. This allows us to capture the continuous and compositional nature of actions, enabling seamless generalization across different environments. Our method extends the capabilities of existing world models by introducing the continuity and compositionality in the form of Lie group structure. We demonstrate the effectiveness of WLA on 2D game and 3D robotics environments in terms of generalization and adaptability.

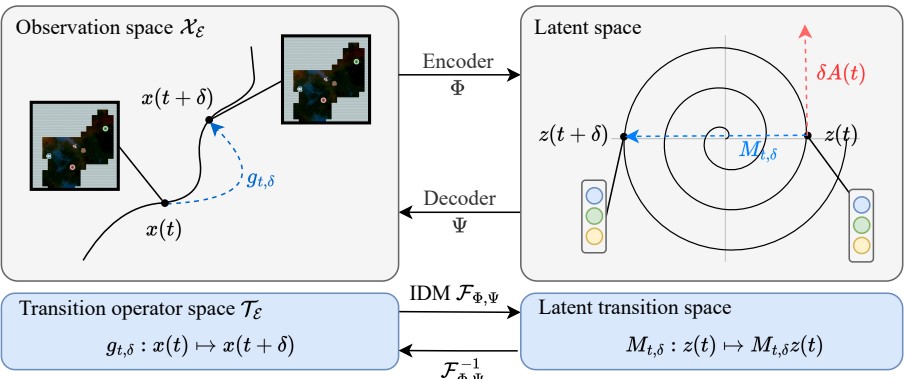

Figure 1: The relation between observation space dynamics and latent space dynamics with Lie group action. We elaborate the design of $\Phi, \Psi$ and the implementation of $\mathcal{F}_{\Phi,\Psi}$ in Section 4.

## 2 PROBLEM SETTING

Before introducing our framework, we formalize the problem that WLA is meant to solve. WLA is a mathematical framework for learning an interface to interact with a family of diverse environments sharing common basic rules of composition and continuity, such as a family of 2D games. To further elaborate, we introduce our definition of *environment*.

### 2.1 ENVIRONMENT

**Definition 1.** *An environment $\mathcal{E}$ is a pair $(\mathcal{X}, \mathcal{T})$ where $\mathcal{X}$ is a space of observations, and $\mathcal{T}$ is a set of nonlinear transition operators on $\mathcal{X}$. Let us use $\mathcal{X}_\mathcal{E}$ and $\mathcal{T}_\mathcal{E}$ to denote the space of observations and the set of transitions of $\mathcal{E}$, respectively. Thus, if $x : I \to \mathcal{X}_\mathcal{E}$ is a sample trajectory of $\mathcal{E}$ with time domain $I \subset \mathbb{R}_+$, then for all $t, t + \delta \in I$, we have $g_{t,\delta}(x(t)) = x(t + \delta)$ for some $g_{t,\delta} \in \mathcal{T}_\mathcal{E}$.*

We assume that the dataset consists of a set of trajectories sampled from all environments $\{\mathcal{E}_j\}_j$. For example, in the family of simple 2D action games, the transitions in one environment may be related to jumping and running. In another environment, actions may contain rotations and and powerups.

Strictly speaking, because $g_{t,\delta}$ differs for each $x$, we should use the notation $g_{t,\delta,x}$ to denote the transition of $x(\cdot)$, and in cases where all trajectories in $\mathcal{E}$ can be enumerated, we should use the notation $g_{t,\delta,i}$ to denote the transition of $x^i(\cdot)$. However, for brevity, we may use $g_{t,\delta}$ in place of $g_{t,\delta,i}$ with the understanding that it is a variable that differs across trajectories. Also, because not all the observation sequences in the real world are realized by a finite set of discrete actions like buttons, we do not assume in this definition that the transitions in the environment can be grounded on such actions (e.g., button presses).

### 2.2 CONTROLLER INTERFACE PROBLEM

We aim to construct an interactive interface for the world model that approximates a given (set of) environment(s), which in this study is/are assumed deterministic. We propose constructing such an interface through an inter-environmental simulator like WLA. In what follows, we rephrase this goal with the definition of environment introduced above. If $\mathcal{A}$ is a topological space of all possible instantaneous inputs to the interface, we informally define the Controller Interface Problem (CIP) for the environment $\mathcal{E}$ as below:

**Problem.** *Using $D_\mathcal{E} = \{x : I \to \mathcal{X}_\mathcal{E}\}$ sampled from $\mathcal{E}$, construct a controller map*

$$Ctrl_\mathcal{E} : \mathcal{X} \times \mathcal{A} \to \mathcal{T}_\mathcal{E} \tag{1}$$

*for which there exists a map $(a : I \to \mathcal{A})$ such that $Ctrl_{\mathcal{E}}(x[0,t), a[0,t))(x(t)) = x(t+dt)$ for all trajectories $x \in D_{\mathcal{E}}$, where $t + dt$ is the infinitesimal future of $t$, and an abuse of notation was used so that $Ctrl_{\mathcal{E}}$ takes as input the time series $(x, a)$ up to $t \in I$, denoted as $x[0,t)$ and $a[0,t)$.* [1]

In more informal words, CIP is a problem of building a controller interface that approximates the rule of dynamics in a target environment. There are two settings for CIP, which we call (1) *unstructured* CIP and (2) *structured* CIP. In *unstructured* CIP, the problem includes that of specifying the space of $\mathcal{A}$ itself. In *structured* CIP, the problem is to find the map with $\mathcal{A}$ that is prespecified, for example, by hardware restrictions. When $\mathcal{A}$ is chosen to be finite, *unstructured* CIP may be analogous to building a set of controller buttons for a game $\mathcal{E}$, and *structured* CIP may be analogous to finding the correspondence map between a given controller input and transitions in the environment. Once the CIP is solved for $\mathcal{E}$, the user may iteratively apply the action sequence of choice to $x \in \mathcal{X}$ to generate a new trajectory within $\mathcal{E}$.

Bruce et al. (2024) solve *unstructured* CIP through a Inverse Dynamic model $f_I : \mathcal{T}_{\mathcal{E}} \to \mathcal{A}$ and a Forward Dynamic model $f_F : \mathcal{X} \times \mathcal{A} \to \mathcal{T}_{\mathcal{E}}$. Meanwhile, $f_F$ is trained so that $f_F(x(t), f_I(g_{t,1})) = x(t+1)$, where the transition $g_{t,1}$ is represented as the pair $(x(t), x(t+1))$. This way, $f_F$ serves as $Ctrl_{\mathcal{E}}$ itself in our formulation of *unstructured* CIP. However, in this paper, we will mainly focus on *structured* CIP. The greatest challenge in *structured* CIP is the deficiency of action labels. We show that WLA can efficiently solve the *structured* CIP with few labels by solving *unstructured* CIP with Lie group action as part of the process. From this point onward, we use "CIP" to imply *structured* CIP unless described otherwise.

# 3 MATHEMATICAL CONCEPT OF WLA FRAMEWORK

WLA is a framework for solving CIP for multiple environments by leveraging the power of an inter-environmental simulator to aggregate the compositional and continuity rules of the dynamics that are shared across all environments. Our inter-environmental simulator uses the algebraic structure of the Lie group and an object-centric autoencoder to represent the continuity and compositionality of the group elements that realize the trajectories $x$, which are sampled across all environments.

Through the Lie-group theoretic structure of transitions learned across multiple environments, WLA solves CIP for all environments by first mapping the input action to the transition operators in the inter-environmental simulator and then by mapping the transitions in the simulator to the observation space to generate the "imagined" trajectory based on the user inputs. The map $Ctrl_{adapt}$ from the action signals to the transitions in the latent space is trained separately. By constructing Ctrl through our inter-environmental simulator instead of the black-box autoregressive model, we can improve the performance of Ctrl on multiple environments.

The core of the WLA framework is the environment-agnostic simulator, which consists of an encoder-decoder pair $(\Phi, \Psi)$ that relates every observation to a state in a vector latent space. While the previous approaches solve CIP separately for each environment, WLA solves the problem through the simulator that describes, in the form of Lie group action, the common compositionality and continuity rules of dynamics in all environments. In the following subsections, we provide an informal explanation to convey the rough idea of each component in our framework. Please see Appendix section A for the formal explanation.

## 3.1 LATENT DYNAMICAL SYSTEM WITH LIE GROUP ACTION

We first describe the mathematical design of our latent space dynamics that ensures the compositionality and continuity of actions. In WLA, we design the transitions to be compositional and continuous in the latent space and train $(\Phi, \Psi)$ so that the transitions in the latent space lift to the observation space while maintaining these properties.

More precisely, if $D = \{x^i : I \to \mathcal{X}\}_i = \bigcup_{\mathcal{E}} D_{\mathcal{E}}$ is a dataset of *time-differentiable paths* in $\mathcal{X}$, we assume that every transition $x^i(t) \to x^i(t+\delta)$ in the trajectory $x^i$ is realized by a group action of a Lie group $G$. That is, for every $x^i(t) \to x^i(t+\delta)$, we assume that there exists $g_{t,\delta}^i \in G$ such that

---

[1] Please see Section A for more formal definition of the controller map.

$g_{t,\delta}^i \cdot x^i(t) = x^i(t+\delta).$ [2] This assumption suits our model-desiderata because the set of $g_{t,\delta}^i$ in this setting can be (i) composed, (ii) inverted, and (iii) differentiated with respect to time ($\delta, t$.)

When a Lie group $G$ acts non-linearly on $\mathcal{X}$, there exists an *equivariant* autoencoder $(\Phi, \Psi)$ such that $G$ acts linearly in the latent space (Koyama et al., 2024). That is, for all $x \in \mathcal{X}$ and $g \in G$,

$$g \cdot x = \Psi(M(g)\Phi(x)) \quad \text{or} \quad \Phi(g \cdot x) = M(g)z(t) \tag{2}$$

where $z = \Phi(x)$ and $M(g)$ is the matrix representation of a Lie group element $g \in G$. To define our inverse dynamic map, we use the following fact that can be readily verified.

**Fact.** *When $g \circ x = \Psi(M(g)\Phi(x))$ for all $g \in G$, then the inverse dynamic map $\mathcal{F}_{\Phi,\Psi} : g \mapsto M(g)$ induced by equation 2 satisfies*

$$\mathcal{F}_{\Phi,\Psi}(h \cdot g) = \mathcal{F}_{\Phi,\Psi}(g) \cdot \mathcal{F}_{\Phi,\Psi}(h), \quad \lim_{\delta \to 0} \mathcal{F}_{\Phi,\Psi}(g_{t,\delta}) = I. \tag{3}$$

This precisely means that our inverse dynamic map $\mathcal{F}_{\Phi,\Psi}$ preserves the compositionality and continuity rules in the observation space. In other words, *the continuity and compositionality of the continuity of the action in the latent space lifts to compositional and continuous action in the observation space through the autoencoder that satisfies equation 2.* See Figure 1 for the schematics of our design that relates the transition in the observation space to the transition in the latent space.

In our design of the transitions $x^i(t) \to x^i(t+\delta) = g_{t,\delta}^i \cdot x^i(t)$ modeled through the latent linear transitions $\Phi(x^i(t)) \to M(g_{t,\delta,i})\Phi(x^i(t))$ in equation 2, we are assuming that the trajectories $x$ are not just continuous but differentiable with respect to $\delta$. Omitting $i$ and writing $M_{t,\delta} := M(g_{t,\delta})$ for brevity, we therefore model $\partial_\delta M_{t,\delta}$ instead of $M_{t,\delta}$ itself. Writing $\partial_\delta M_{t,\delta}|_{\delta=0}$ as $A(t)$ and $z(t) = \Phi(x(t))$, we eventually arrive at the following latent linear dynamical system

$$\frac{d}{dt}z(t) = A(t)z(t), \quad \text{or} \quad z(t) = \exp\left(\int_0^t A(s)ds\right)z(0). \tag{4}$$

As a part of our design, we also assume that $G$ in question belongs to a well-known family of groups whose matrix representations are scalings and rotations. For the group of rotation and scaling, every $A(s) = \bigoplus_k A_k(s)$ and $M_{t,\delta} = \bigoplus_k M_{k,t,\delta}$ are respectively direct sums of the matrices of the form

$$A_k(s) = \begin{pmatrix} \lambda_k(s) & -\theta_k(s) \\ \theta_k(s) & \lambda_k(s) \end{pmatrix}, \quad M_{k,t,\delta} = \exp(\Lambda_k(t,\delta))\begin{pmatrix} \cos\Theta_k(t,\delta) & -\sin\Theta_k(t,\delta) \\ \sin\Theta_k(t,\delta) & \cos\Theta_k(t,\delta) \end{pmatrix} \tag{5}$$

where $\Lambda_k(t,\delta) = \int_t^{t+\delta} \lambda_k(s)ds$, $\Theta_k(t,\delta) = \int_t^{t+\delta} \theta_k(s)ds$.

### 3.2 OBJECT CENTRIC DYNAMICS MODEL

To introduce the notion of objects into our inter-environmental simulator, we adopt the idea of object-centric modeling and train $(\Phi, \Psi)$ with a latent space partitioned into distinct objects. In particular, we design the encoder $\Phi$ to map $x$ into a set of $N$ slots $[z_n]_n^N$ so that the latent space $\mathcal{Z}$ is partitioned into slot spaces $\mathcal{Z}_n$. Because the transitions happen for each slot, $A_k$ in equation 5 are thus enumerated by both the number of objects $N$ and number of rotation angles $J$, so that $[A_k]_k^J = [A_{nj}]_{n,j}^{N,J}$ by reindexing, and $\mathcal{Z}_n = [\mathcal{Z}_{nj}]_j^J$. Analogously, let us write $\Phi_n = [\Phi_{nj}]_n^N$. Likewise, we re-index $(\lambda_k, \theta_k)$ as $(\lambda_{nj}, \theta_{nj})$ as well.

### 3.3 USING WLA TO SOLVE CIP

Technically, just like how humans can "imagine" the trajectory in our abstract mental model without buttons and levers, we can use the rotation parameters and scaling parameters in our trained simulator as $\mathcal{A}$ itself in the setting of CIP. However, in this section, we consider the problem of leveraging $(\Phi, \Psi)$ to solve CIP with pre-specified $\mathcal{A}$. Such a situation may arise in robotics, for example.

As we have described in the previous section, once $(\Phi, \Psi)$ are obtained in a way that satisfies equation 2 on multiple environments, we have the inverse dynamic map (IDM), $\mathcal{F}_{\Phi,\Psi}$, along with its

---

[2]Note that we wrote $g \cdot (x)$ instead of $g(x)$ to designate a group action. Please see A for formal discussion.

Figure 2: WLA is based on a slot-attention-based autoencoder. The latent space is partitioned into slots, and the transition for each slot occurs by a linear Lie group action. (a) During the training of WLA, the inverse dynamic map $\mathcal{F}_{\Phi,\Psi}$ converts the transition to a Lie algebra operator. (b) The forward rollout simulation with a controller interface is implemented by mapping the contextual inputs (past observations and external action signals) to Lie algebra parameters $(\lambda, \theta)$ and by multiplying the resulting operators $M$ to the slot token $z[t]$ to create the future observations autoregressively.

inverse $\mathcal{F}_{\Phi,\Psi}^{-1}$ in equation 3. When there is a user-specified action signal space $\mathcal{A}$, we can use $\mathcal{F}_{\Phi,\Psi}^{-1} : \mathcal{A} \to \mathcal{T}$ as an intermediate interface in our construction of $\mathrm{Ctrl}_{\mathcal{E}}$.

More specifically, given the action labeled sequence $[(x(t), a(t))]_t$ with $(x(t), a(t)) \in \mathcal{X}_{\mathcal{E}} \times \mathcal{A}$, we can learn a map $\mathrm{Ctrl}_{\mathcal{E}}$ by training a model $\mathrm{Ctrl}_{adapt}$ that takes $(x[0, t), a[0, t))$ as the input and outputs $(\lambda(t), \theta(t))$ which are the parameters of $A(t) := A(\lambda(t), \theta(t))$, where $[\lambda]_k = \lambda_k$ and $[\theta]_k = \theta_k$ are as in equation 5. This way, we construct a controller map by composing $\mathrm{Ctrl}_{adapt}$ with the $\mathcal{F}_{\Phi,\Psi}^{-1}$. That is, we can construct the map $\mathrm{Ctrl}_{\mathcal{E}} : \mathcal{X} \times \mathcal{A} \to \mathcal{T}_{\mathcal{E}}$ via the composition

$$\mathrm{Ctrl}_{\mathcal{E}} : x[0, t), a[0, t) \underset{\mathrm{Ctrl}_{adapt}}{\longrightarrow} A(\lambda(t), \theta(t)) \underset{\mathcal{F}_{\Phi,\Psi}^{-1}}{\longrightarrow} M_{t,\delta} \tag{6}$$

where $\delta$ is the chosen time-discretization stepsize in approximating the continuous dynamics numerically so that the $\lambda(t)$ and $\theta(t)$ are assumed to take the same value over the interval $[t, \delta)$. The output of $\mathrm{Ctrl}_{\mathcal{E}}$ in the form of $M_{t,\delta}$ realizes the transition $x(t) \to x(t + \delta)$ through $x(t + \delta) = \Psi(M_{t,\delta}\Phi(x(t)))$. We describe the learning scheme of $\mathrm{Ctrl}_{adapt}$ in the next section.

## 4 IMPLEMENTATION OF WLA MODEL

We use neural networks for the modeling of all components of WLA, including Ctrl and $(\Phi, \Psi)$. In this section, we first remark on our practical handling of the dataset and then explain the training loss and our architectural design choices. Also see Figure 2 along with the materials in this section.

### 4.1 TIME DISCRETIZATION

In an implementation, the time series dataset is discretized, and an interval $I$ is represented as an increasing subset $\{\tau_0 < \tau_1 < \cdots < \tau_T\} \subset R_+$. As such, a trajectory $x(\cdot)$ is represented by the discrete signal $[x(\tau_t)]_t$. From here on forward, whenever we use the "hard bracket" $x[t] := x(\tau_t)$, we assume that the signal is discretized and that $t$ is a natural number. Also, just for brevity, we assume in this section that the observations are evenly spaced, so that $\tau_t - \tau_{t-1} = \Delta$. We also denote the past observations as $x[: t] := (x[0], \ldots, x[t-1])$. We can similarly define the discretized version of other variables such as $a[t], A[t], \lambda[t], \theta[t]$.

In CIP, $\mathrm{Ctrl}_{\mathcal{E}}$ was designed to take $(x[: t], a[: t])$ to compute $\lambda_k[t]$ and $\theta_k[t]$. Likewise, we implement IDM $\mathcal{F}_{\Phi,\Psi}$ as a function that takes two consecutive frames $(x[t], x[t + 1])$ as an input instead of the actual transition $g_t$, because the operator itself is not observable. This strategy has been successfuly taken in the past (Mitchel et al., 2024; Koyama et al., 2024; Miyato et al., 2022). Because this section pertains to implementation, we will use this discretized notation throughout this section. See Appendix A for a more formal explanation.

## 4.2 TRAINING OF $(\Phi, \Psi)$

The WLA model is trained in an unsupervised manner using a set of trajectories sampled from different environments. We use an approach similar to (Koyama et al., 2024; Mitchel et al., 2024) except that because we are modeling a time series whose velocity changes with respect to time, we use the prediction loss over the entire time sequence. Namely, if $D$ is the dataset of trajectories, we train the autoencoder $\Phi, \Psi$ and the IDM $\mathcal{F}_{\Phi,\Psi}$ along with the trainable parameters $\{\lambda_{nj}[t], \theta_{nj}[t]\}$ of $\{A_{nj}[t]\}$ in the discretized version of equation 5 (defined for each $x \in D$ and time $t$ with the reindexing of $k$ with $nj$ as in 3.2 [3]) by minimizing

$$\sum_{x \in D} \mathcal{L}_{\text{fwd}}(x) + \mathcal{L}_{\text{bwd}}(x) + \alpha \mathcal{L}_1(\lambda, \theta) \tag{7}$$

$$\text{where} \quad \mathcal{L}_{\text{fwd}}(x) = \sum_t ||x[t] - \hat{x}_f[t]||^2, \quad \mathcal{L}_{\text{bwd}}(x) = \sum_t ||x[t] - \hat{x}_b[t]||^2, \tag{8}$$

in which the forward and backward predictions $\hat{x}_f, \hat{x}_b$ are given by

$$\hat{x}_f[t] = \Psi\left(\exp\left(\Delta \sum_{0 \le \ell < t} A[\ell]\right) z(0)\right), \quad \hat{x}_b[t] = \Psi\left(\exp\left(-\Delta \sum_{t \le \ell < T} A[\ell]\right) z[T]\right), \tag{9}$$

where the summation inside the exponential is derived from modeling $A(s)$ discretely as $A[t] \cdot 1_{s \in [\tau_t, \tau_{t+1})}$ so that $\int_0^{\tau_t} A(s)ds = \Delta \sum_{\ell < t} A[\ell]$. Finally, we used the sparsity loss $\mathcal{L}_1 = \sum_{j,n} |\lambda_{nj}[\ell]| + \sum |\theta_{nj}[\ell]|$. Once $(\Phi, \Psi)$ is solved, CIP provides its own controller maps defined by $\text{Ctrl}(g_{\delta,t})$.

## 4.3 SOLVING CIP THROUGH WLA

To learn $\text{Ctrl}_{adapt}(x[t], a[t]) \mapsto A(\lambda[t], \theta[t])$ (the time-discretized version of 3.3), we use a labeled dataset $\{(x[t], a[t]\}$ of sequences paired with action inputs. We minimize the sum of the following adaptation loss and reconstruction loss, defined for all $t, t_0$ and $x$ as

$$\mathcal{L}_{\text{adapt}} = ||[\lambda(t), \theta(t)] - \text{Ctrl}_{adapt}(x[:t], a[:t])||^2, \quad \mathcal{L}_{\text{rec}}(x; t, t_0) := ||x[t] - \hat{x}[t \mid t_0]||^2 \tag{10}$$

where $\hat{x}[t \mid t_0]$ is a rollout prediction from $x[t_0]$, constructed by iteratively applying $a[t]$.

## 4.4 ARCHITECTURE

For the encoder-decoder pair $(\Phi, \Psi)$, we used a Transformer-based model with the slot attention mechanism (Locatello et al., 2020). We employed the Vision Transformer (ViT)-based slot encoder (Wu et al., 2023) so that the slot attention module in the encoder partitions the input $x$ into a set of slots $[z_n]_n^N$ through the cross-attention architecture with a learnable set of initial slot tokens. To obtain the output, we used a standard ViT decoder which (1) first copies each slot token $z_n$ as the initial values of patch tokens $P_n := [p_{n,l}]_l$ and then (2) computes the RGB image $u_n$ and the alpha mask $w_n \in [0, 1]^{H \times W}$ of spatial size $H \times W$ via self-attention mechanisms over $P_n$. The decoder finally outputs $\hat{x}$ as the weighted mean $\sum_n w_n * u_n$ where $*$ is the element-wise product. For the IDM, we use an MLP that takes the concatenation of the slots $z_n[t] \oplus z_n[t+1]$ and outputs both $\lambda_{n,j}[t]$ and $\theta_{n,j}[t]$.

There are two important hyperparameters in our modeling: the number of slots $N$ and the number of Lie group actions $J$. Increasing $N$ and $J$ generally improves performance but also increases computational complexity. We will describe more details in Appendix.

**Slot Alignment via least action principle** A naive application of the slot attention mechanism to dynamical scenarios tends to suffer from temporal inconsistencies, i.e., slots fail to track objects consistently over time (Zhao et al., 2023). To encourage temporal consistency of object slot assignments, we introduced the principle of least action (Siburg, 2004). Given the next and current frames $z_n[t+1], z_n[t] \in \mathcal{Z}_n$ for all slots $n = 1, \ldots, N$, we chose the permutation $\sigma$ to the slots so that

---

[3]These parameters are 'not' to be stored as parts of the model.

the transition $z_n[t+1] \to z_{\sigma(n)}[t+1]$ in the latent space is minimal. Instead of computing $A$ indexed by slots, we thus computed $[A_{n_1 \to n_2}]^{N \times N}_{n_1, n_2}$ and used a linear assignment problem solver for the permutation $\sigma$ that minimizes $\|A_{n_1 \to \sigma(n_1)}\|^2$, based on the implementation in Karpukhin & Savchenko (2024)

## 5 RELATED WORK

Several works have been done on (latent) dynamics models that can generalize across environments, many of them purporting to solve a reinforcement learning problem. Hafner et al. (2023) propose a method designed for the purpose of solving reinforcement learning tasks in various environments and uses a recurrent latent space and discretized states learned with action and reward-labeled sequences. Lee et al. (2022) also use multiple environments to train generalist reinforcement learning agents. Our method, on the other hand, is an unsupervised generative interactive framework trained to generate a video itself in response to user action inputs. Raad et al. (2024) use a language-annotated dataset to build a world model that can be used to solve tasks in multiple environments. Mondal et al. (2022) associate group actions with the transition as in our approach but focus on embedding-invariant reward optimization and do not generate future observations.

The three major features of WLA are that it is (1) unsupervised, (2) object-centric, and (3) able to capture the continuous dynamics in a way that all transitions are compositional by design. We borrow much of the idea from (Wu et al., 2022), an object-centric framework with slot attention architecture and temporal position embedding. It preserves the temporal consistency through residual connections, in contrast to our algebraic structure. To encourage the compositionality of dynamics, Rybkin et al. (2018) combine past frames through an MLP to predict future frames. Valevski et al. (2024) use conditional diffusion to model the dynamics autoregressively, providing a high-quality continuous simulator. Because the effect of the action is modeled in a black box, the input of "no action" does not necessarily map to an identity operator in the observation space, which corrupts the generation in the long run. To mitigate this undesired effect, they use the technical trick of noise augmentation. For the algebraic design of our Controller map that guarantees the compositionality, we build on the theory of (Koyama et al., 2024; Mitchel et al., 2024), which are unsupervised methods that learn the underlying symmetry structure by predictions. Their method, however, does not consider continuous dynamics that are non-autonomous (time-dependent). Mondal et al. (2024) consider Koopman-based modeling with complex diagonal latent transition operators related to our $\mathcal{A}$, but their latent actions are additive, and ambient transitions are time-homogeneous.

As an effort to learn an interactive generative environment, (Watter et al., 2015) have advocated to control dynamical systems in latent space. However, our method has much more in common with VPT (Baker et al., 2022). VPT learns $F_I : \mathcal{X} \times \mathcal{T}_\mathcal{E} \to \mathcal{A}$ and $\text{Ctrl}_F : \mathcal{X} \times \mathcal{A} \to \mathcal{T}_\mathcal{E}$ separately as noncausal and causal maps, and it learns the noncausal part with action labels. WLA is different in that it learns without labels the $\mathcal{E}$-agnostic noncausal structure in the form of an underlying Lie group via equivariant autoencoder $(\Phi, \Psi)$. Genie (Bruce et al., 2024) and LAPO (Schmidt & Jiang, 2023) also share a similar philosophy as VPT, except that they do not use labels in learning $\text{Ctrl}_I$. Unlike WLA, their controller interface does not map actions through the latent space that reflects the structure of compositional and continuous action. (Goyal et al., 2021) took the approach of separating *object files* from *schemata* to describe the dynamics. However, in their context, our schemata is modeled with much stronger inductive bias of compositionality and continuity. Chen et al. (2023) build an interactive system by conditioning the diffusion generation on user stroke inputs. However, they do not consider the dynamics behind the system with objects and compositionality.

## 6 EXPERIMENTS

### 6.1 PHYRE

The Phyre benchmark provides a set of physical reasoning tasks in 2D simulated environments. We used it as a sanity check to validate that our model possesses the capabilities of continuity and compositionality of actions. By focusing on physical interactions that require understanding continuous dynamics and the compositional nature of actions (e.g., applying forces, combining movements), Phyre is an ideal platform to assess these fundamental aspects of our model.

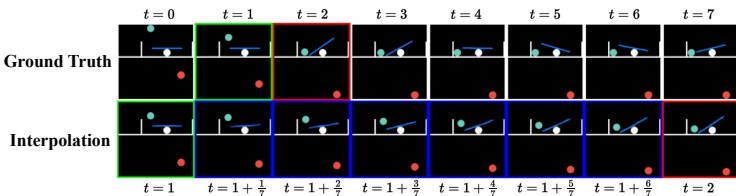

Figure 3: Phyre Interpolation. **Top**: Training frames at a low sampling rate (1 FPS). **Bottom**: Reconstructed trajectory at a high sampling rate (8 FPS), with interpolated frames shown in blue.

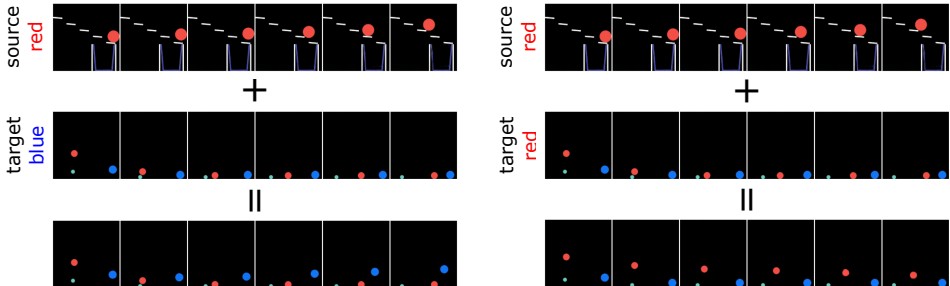

Figure 4: Composition results on Phyre. **Left**: Applying the sum of actions from the red and blue ball to the blue ball. Since the red ball is climbing, its action counteracts the falling action of the blue ball. **Right**: Applying the sum of actions to the red ball, showing similar compositional effects.

**Results:**     Figure 3 demonstrates the interpolation capabilities of our model on Phyre. During training, the model observes frames at a low sampling rate (1 FPS). We tested the model's ability to recover trajectories at a higher sampling rate (8 FPS) by generating interpolated frames between observed frames. The interpolated frames, displayed with blue padding, illustrate the model's understanding of continuous dynamics and its ability to generate smooth transitions, validating the continuity aspect. In Figure 4, we showcase the compositionality of actions learned by our model. By applying combinations of actions to objects and observing the effects (e.g., counteracting movements), we demonstrate that the model effectively captures the compositional nature of actions within the environment. These results validate that WLA has the capabilities of continuity and compositionality of actions that are essential for simulating physical dynamics in complex environments.

## 6.2 PROCGEN

The ProcGen benchmark offers a suite of procedurally generated game-like environments, providing diverse challenges for testing generalization and adaptability. We used datasets provided by Schmidt & Jiang (2023). In this benchmark, we addressed the Controller Interface Problem for both in-domain (seen) and out-of-domain (unseen) datasets, with and without action labels, referred to as in-play and out-of-play settings, respectively. Importantly, in all experiments, we trained and evaluated a *single common model* across all environments.

We compared our method specifically against Genie (Bruce et al., 2024). Although Genie is designed to function without action labels, to evaluate it for the Controller Interface Problem with labels, we incorporated trainable embeddings of action labels and appended them to the output of the action embeddings in their latent action model (LAM). We utilized the open-source Genie codebase implementation provided by Willi et al. with default settings. However, we increased the number of training iterations from 0.2M to 0.4M to accommodate our multi-environment as opposed to the original work, which trained separate models for different environments.

**Evaluation Metrics:**     We used the following metrics to evaluate our methods (i) PSNR (Peak Signal-to-Noise Ratio), (ii) $\Delta_t$ PSNR, (iii) LPIPS (Learned Perceptual Image Patch Similarity) (Zhang et al., 2018), and (iv) ActionACC (Action Accuracy). $\Delta_t$ PSNR is a metric introduced by Bruce et al. (2024) that measures the difference between the effect of the inferred ac-

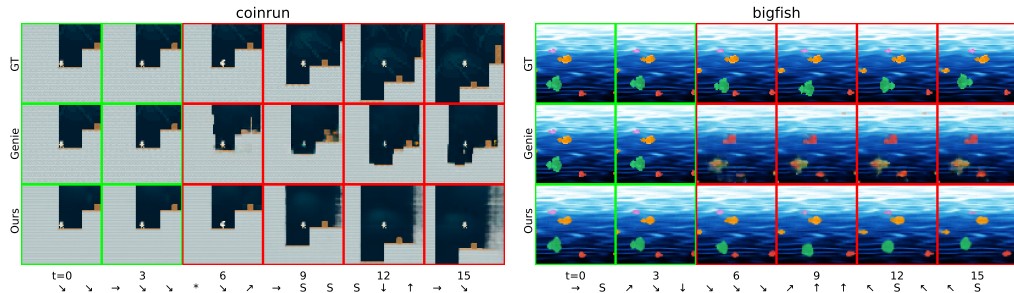

Figure 5: The controlling results through $\text{Ctrl}_{adapt}$ on ProcGen. The figure contains 6 out of 16 frames, resulting from applying the action sequence written below the rendered frames.

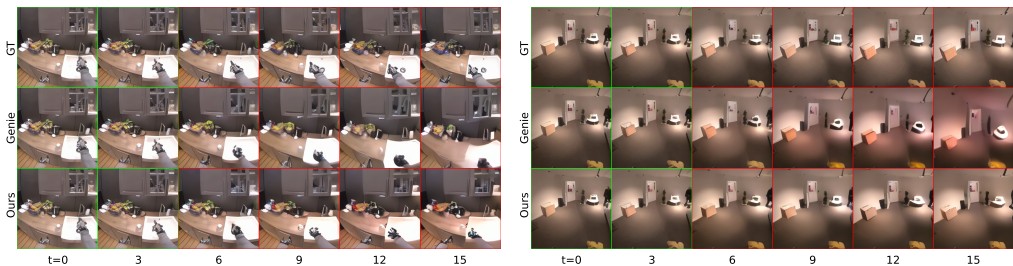

Figure 6: The controlling results through $\text{Ctrl}_{adapt}$ on Android Dataset.

tion and the effect of the random action. Specifically, $\Delta_t = \text{PSNR}(x_{t+1}, \text{Ctrl}_{adapt}(x(t), a(t)) - \text{PSNR}(x_{t+1}, \text{Ctrl}_{adapt}(x(t), a_{random}(t))$. ActionACC measures the accuracy of the estimated actions compared to the ground truth action labels. To estimate the actions, we trained a logistic linear regressor that maps the model-inferred parameters $(\lambda(t), \theta(t))$ to the ground truth action labels $a(t)$ and evaluated the label accuracy. For Genie, we similarly applied the logistic regression to the inferred action tokens provided by LAM.

**Results:** The results of our experiments on seen environments are summarized in Table 2. Our method consistently outperformed Genie across multiple metrics. Figure 5 illustrates the results of applying controller inputs to various game environments via $\text{Ctrl}_{adapt}$. Each sequence shows 6 out of 16 frames resulting from applying the corresponding action inputs. The visualizations demonstrate that our model can generate coherent and responsive sequences. Please also see the supplementary material for the movies generated using our method.

**Ablation Study:** We conducted ablation studies to assess the impact of two key components of WLA. (i) Lie Group Action (Rotation): We tested a version of our model without the rotational components in the Lie group transitions in equation 4, i.e., using only scaling transformations. (ii) Least Action Principle for Slot Alignment: We evaluated the model without using the least action principle for aligning slots over time. The results, presented in Table 1, demonstrate that both components significantly contribute to the model's performance. Removing the rotational components adversely

| | MSE($\downarrow$) | |
| --- | --- | --- |
| | Unseen | Seen |
| w/o Rotation | 0.683 | 0.059 |
| w/o Least Action | 0.675 | 0.056 |
| Ours | **0.602** | **0.046** |

| | ActionACC($\uparrow$) | |
| --- | --- | --- |
| | Unseen | Seen |
| Genie | 8.30 | 10.25 |
| Ours | **14.62** | **21.07** |

Table 1: **Left**: ablation study for the effect of Lie group action and the least action principle. **Right**: Action accuracy in the out-play setting.

| Environ. | PSNR($\uparrow$) | | $\Delta_t$PSNR($\uparrow$) | | LPIPS($\downarrow$) | |
|---|---|---|---|---|---|---|
| | Genie | Ours | Genie | Ours | Genie | Ours |
| bigfish | 18.69 | **24.04** | -0.09 | **1.26** | 0.14 | **0.04** |
| bossfight | 15.76 | **18.48** | 0.02 | **0.29** | **0.18** | 0.19 |
| caveflyer | 11.25 | **17.59** | 0.02 | **2.45** | 0.22 | **0.18** |
| climber | 15.22 | **19.41** | 0.03 | **2.68** | 0.21 | **0.18** |
| coinrun | 11.30 | **22.10** | 0.48 | **9.03** | 0.21 | **0.05** |
| maze | 18.44 | **21.68** | -0.24 | **1.52** | 0.20 | **0.13** |
| miner | 18.85 | **21.75** | -0.06 | **1.19** | 0.11 | **0.09** |
| ninja | 13.46 | **19.63** | 0.05 | **4.06** | 0.32 | **0.18** |

Table 2: Summary of the simulation performance through Ctrl$_{adapt}$ in the seen setting on ProcGen.

| | Genie | Ours |
|---|---|---|
| PSNR($\uparrow$) | **21.16** | 20.82 |
| $\Delta_t$PSNR($\uparrow$) | 0.78 | **1.13** |
| FVD($\downarrow$) | 393.85 | **131.02** |

Table 3: Quantitative Results on the android dataset.

affected the model's ability to learn dynamics across diverse environments. Note that the ablated version (w/o rotation) is similar to Mamba (Gu & Dao, 2023), which uses a state space model with diagonal action. Similarly, omitting the least action principle decreased object-wise consistency, which is crucial for maintaining our model's compositional and continuous dynamics.

## 6.3 ANDROID DATASET

To further validate our method, we compared WLA against Genie on the 1X World Model dataset[4]. This dataset comprises over 100 hours of videos capturing the actions of android-type robots in various environments and lighting conditions, including narrow hallways, spacious workbenches, and tabletops with multiple objects. We slightly adapted the architecture of our method to suit this setting, as detailed below, but otherwise, we followed the same experimental protocol as the ProcGen dataset. We evaluated the models using three metrics: PSNR, $\Delta_t$PSNR, and Fréchet Video Distance (FVD) (Unterthiner et al., 2018), utilizing the debiased version (Ge et al., 2024).

**Results:** We observed results that were consistent with those from the ProcGen dataset. While Genie produces cleaner predictions for individual frames, it falls behind WLA in generating video sequences that align with the provided action sequences. In contrast, our method successfully learns the correct responses, producing videos more closely resembling the ground truth (Figure 6). These qualitative observations are supported by the quantitative results shown in Table 3. Although our method slightly underperforms Genie in frame-wise evaluation (PSNR), it performs better temporally local evaluation ($\Delta_t$PSNR) and significantly outperforms Genie in FVD. Some of the predicted videos are included in the supplementary material.

## 7 CONCLUSION

In this paper, we introduced WLA for the CIP, a method that uses a simulator to capture the rules of composition and continuity between environments through the algebraic relations of a Lie group and object-centric modeling. Using the structured latent model, we can improve the controllability and predictive power of the controller interface. The empirical results demonstrate that WLA is capable of modeling real-world robot actions in 3D environments, in addition to 2D game environments. Importantly, it is the first of its kind as a generative interactive framework that is based on a state-space model. However, there are still several limitations that need to be resolved to scale up the framework further. Firstly, our method does not account for the possible randomness of the environment. This problem might be addressed by utilizing stochastic process modeling, for example. Secondly, in our model, we assume a priori that transitions in the environment commute with each other, and the number of rotations in the latent dynamics is specified by the user. NFT (Koyama et al., 2024), one source of our inspiration, however, learns the group dynamics without relying on such an assumption. Future work, therefore, includes the development of methods to build a latent state-space model with fewer prior assumptions and more freedom.

---

[4]Available at `https://huggingface.co/datasets/1x-technologies/worldmodel`; we used version 1.1.

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

# A   MORE FORMAL SETUP OF WLA AND CIP PROBLEM

In this section, we provide a more formal version of the mathematical setup of our framework, which is mostly built on the theory of NFT (Koyama et al., 2024), except that we use continuous and nonstationary time series for training the model. We first provide the list of notations, and elaborate the details in the following order: (1) Dataset, (2) Model, (3) CIP, and (4) practical modification of the setups for implmentation purposes.

## A.1   NOTATIONS

- $\mathcal{X}$ ; the space of observations, assumed to be some smooth manifold.

- $\mathcal{X}_{\mathcal{E}} \subset \mathcal{X}$ ; the space of observations for environment $\mathcal{E}$.

- $G$ ; Lie Group.

- $I$ ; Time domain, can be assumed to be a compact subset of $\mathbb{R}_+$.

- $V$ ; $D$ dimensional latent vector space over the field $\mathbb{R}$.

- $\mathcal{A}$ ; a topological space containing the set of 'action' signals.

- $T_h\mathcal{M}$ ; a tangent space of a manifold $\mathcal{M}$ at $h \in \mathcal{M}$.

- $C(X, Y)$ ; a continuous map from a topological space $X$ to $Y$.

- $C_k(X, Y)$ ; a $k$-differentiable map from a smooth manifold $X$ to $Y$.

- $\tilde{J}$ ; number of toric component in $G$

- $D$ ; the dimension of the latent space.

- $F$ ; number of irreducible representations / frequencies in the representation of $G$ used in our work

- $h(X)$ ; a subset $\{h(x) \mid x \in X\} \subset Y$

- $\Phi$ ; an invertible encoder map in $C_{\infty}(\mathcal{X}, V)$

- $\mathrm{GL}(V)$ : General linear group of $V$.

- $\mathrm{gl}(V)$ : Linear endomorphism of $V$.

- $\Psi$ ; the inverse map of $\Phi$, an element of $C_{\infty}(V, \mathcal{X})$.

- $D_{\mathcal{E}}$ ; an $\mathcal{E}$-labeled subset of $C(I \to \mathcal{X}_{\mathcal{E}})$

- $\mathcal{T}_{\mathcal{E}}$: a set of transition operators $\tau : \mathcal{X} \to \mathcal{X}$ such that whenever $x \in D_{\mathcal{E}}$ and $t > 0, \delta > 0$, there exists some $\tau \in \mathcal{T}_{\mathcal{E}}$ such that $x(t + \delta) = \tau(x(t))$.

- $g \cdot x$ ; an abbreviation of $\alpha(g, x)$ when $\alpha : \mathcal{G} \times \mathcal{X} \to \mathcal{X}$ is a group action.

- $\exp_G$ ; exponential map for Lie group $G$. We omit the suffix when it is self-implied.

- $h[0, t)$ ; $\{h(s) | s \in [0, t)\}$.

## A.2 Assumptions of Dataset

Let $D = \{x^i\}_i \subset C_1(I, \mathcal{X})$ be a video dataset to be used to train the encoder $\Phi$ and the decoder $\Psi$ in our WLA framework, where $i$ is the sample index. This dataset contains the videos sampled from multiple environments so $\mathcal{X}_\mathcal{E} \subset \mathcal{X}$ for all $\mathcal{E}$. The underlying assumption of WLA is that, for every $x \in D$, $t > 0$ and $\delta > 0$, the transition $x(t) \to x(t + \delta)$ is *realized* by *some* common group action $G \times \mathcal{X} \to \mathcal{X}$ of a *some* Lie group $G$. To be more precise, our standing assumption is as follows:

**Assumption 1.** *For our dataset $D = \{x^i : I \to \mathcal{X}\}_i$ consisting of movie samples from multiple environments, there exists some (abelian) Lie group $G$ that acts on $\mathcal{X}$ through a group action $\alpha :$ $G \times \mathcal{X} \to \mathcal{X}$ such that, for every $x^i \in D$, $t > 0$ and $\delta > 0$ there is some $g^i_{t,\delta} \in G$ such that $\alpha(g^i_{t,\delta}, x^i(t)) = x^i(t + \delta)$.*

For brevity, we notationaly identified $g$ with the map $\alpha_g : x \mapsto \alpha(g, x)$ (that is, we interpret $g^i_{t,\delta} : \mathcal{X} \to \mathcal{X}$ with $g^i_{t,\delta}(x^i(t)) = x^i(t + \delta)$. ) With this identification, we are therefore assuming that the family of transitions $T_\mathcal{E}$ can be identified with a subset of $G$ for every $\mathcal{E}$. We introduce this assumption of the Lie group structure as our inductive bias because our desiderata of composition-ality and continuity matches (almost exactly, except for the invertibility) with the definition of Lie group as a topological set with smooth manifold structure and rules of compositions.

For the computational reasons that we will elaborate on momentarily, we also make an additional assumption on $G$, that is, (1) $G$ is connected and compact so that the exponential map is surjective on $G$ and (2) $G$ is abelian. Although these restrictions may seem strong (especially the abelian assumption), we will practically resolve this problem through non-autonomous modeling of the time series.

## A.3 Latent Space Modeling with Lie Action

According to the theory and results presented in NFT, when there is some group $G$ acting on $\mathcal{X}$ with a set of reasonable regularity conditions, there exists an autoencoder $(\Phi, \Psi)$ with latent vector space $V$ and a homomorphic map $M : G \to \mathrm{GL}(V)$ such that $\Phi(M(g)\Psi(x)) = g(x)$. This $M$ is also referred to as *representation of $G$*. Mitchel et al. (2024) and Miyato et al. (2022) had experimentally shown that such $(\Phi, \Psi)$ and $M$ can be learned from a time series dataset, and our work scales their philosophy to the non-stationary, continuous dynamical system on large observation space.

To model the non-autonomous time series we assume that, for all $x \in D$ there exists a time series $g(\cdot) \in C(I, G)$ such that $x(t) = g(t) \cdot x(0)$. This way, we would have

$$x(t) = \Psi\left(M(g(t))\Phi(x_0)\right).$$

Because we want to model the **continuous** evolution of $x$ forward in time, we model its derivative instead of modeling $g(t)$ for each $t$. By our assumption on the surjectivity of the exponential map, every $g(t)$ may be written as $\exp_G(\mathfrak{g}(t))$, where $\mathfrak{g} \in T_{id}G$ is a tangent vector of $G$ at the identity element $id \in G$, i.e., the Lie algebra of $G$. By the abelian assumption we can WLOG write

$$\mathfrak{g}(t) = \int_0^t \mathfrak{a}(s)ds$$

for $\mathfrak{a} \in C(I, T_{id}G)$. Writing $\tilde{M} : T_{id}G \to gl(V)$ to be the Lie algebra representation associ-ated with $G$ so that $\exp_{\mathrm{GL}(V)} \circ \tilde{M} = M \circ \exp_G$, we arrive at the following continuous time series dynamics through latent linear Lie action on $V \supset \Phi(\mathcal{X})$;

$$x(t) = \Psi\left(M\left(\exp\left(\int_0^t (\mathfrak{a}(s))ds\right)\right)\Phi(x(0))\right) = \Psi\left(\exp\left(\int_0^t \tilde{M}(\mathfrak{a}(s))ds\right)\Phi(x(0))\right) \quad (11)$$

$$:= \Psi\left(\exp\left(\int_0^t A(s)ds\right)\Phi(x(0))\right) \quad (12)$$

where we denoted $\tilde{M}(\mathfrak{a}(s)) = A(s)$, omitting the suffixes of $\exp$ for brevity. Altogether, we write

$$x(t) = \Psi\left(\exp\left(\int_0^t A(s)ds\right)\Phi(x(0))\right) \quad (13)$$

**An remark Regarding the notation of $\mathbf{g_{t,\delta}}$ :** In alignment with the $g_{t,\delta}$ notation we introduced in the A.2, an instance $g_{t,\delta} \in G$ that realizes the transition $x(t) \mapsto x(t + \delta)$ may be written as $g_{t,\delta} = \exp_G \left( \int_t^{t+\delta} \mathfrak{a}(s)ds \right)$. Because we have tools in this Appendix section to elaborate our design of continuous transition with integration, however, we do not use this notation in the subsequent formal explanations of CIP.

## A.4 CONTROLLER MAP

For the environment with Controller interface, we assume that there exists an environment-specific, continuous Controller map

$$\mathrm{Ctrl}_{\mathcal{E}} : C(I, \mathcal{X}) \times C(I, \mathcal{A}) \times I \to T_{id}G$$

along with a possibly small paired dataset $D_{adapt} \subset C_1(I, \mathcal{X}) \times C(I, A)$ such that, whenever $(x, a) \in D_{adapt}$, we have

$$x(t) = \Psi \left( \exp \left( \int_0^t \tilde{M}(\mathrm{Ctrl}_{\mathcal{E}}(x[0, s], a[0, s)))ds \right) \Phi(x(0)) \right) \tag{14}$$

where we wrote $\mathrm{Ctrl}_{\mathcal{E}}(x[0, s], a[0, s))) := \mathrm{Ctrl}_{\mathcal{E}}(x, a, s)$ to convey the contraint that the controller output at $s$ depends only on the history of $x$ and $a$ up to $s$. Indeed, the purpose of CIP is to train $\mathrm{Ctrl}_{\mathcal{E}}$ based on $D_{adapt}$. Because the input to $\mathrm{Ctrl}_{\mathcal{E}}$ is not $x(s), a(s)$, the controller map $\mathrm{Ctrl}_{\mathcal{E}}$ is expected to entail the causal mechanism that cannot be captured with $(\Phi, \Psi)$ alone.

## A.5 PRACTICAL DETAILS FOR THE PURPOSE OF IMPLEMENTATION

Although we defined the formal output of $\mathrm{Ctrl}_{\mathcal{E}}$ to be $T_{id}G$ , the Lie algebra $T_{id}G$ itself is an abstract entity. To instantiate $T_{id}G$ in numerical form, we use the matrix representation $\tilde{M} : T_{id}G \to \mathrm{gl}(V)$ of Lie algebra. Also, we use the fact that a connected abelian Lie group is isomorphic to $\mathbb{R}^{J_1} \times \mathbb{T}^{J_2}$ so that any element of $G$ can be characterized by a set of pairs of scale $\lambda$ and rotation angle $\theta$. Likewise, an element of corresponding Lie algebra $\mathfrak{g}$ can be characterized by a set of pairs of scaling speed and rotational speed. This way, we can conclude that the matrix representations $\tilde{M}(\mathfrak{g})$ of our $T_{id}G$ are parametrized by a set of scale-rotation velocity pairs and irreducible representations (frequencies). In this work, we assume $J_1 = J_2 = \tilde{J}$ for brevity and denote it so. [5]

Reindexing these pairs of *rotation and scale* with the frequencies $\{r_f | f = 1, ...F\} \subset R$ (irreducible representation) and the index of toric component $\{1, ..., \tilde{J}\}$, let us therefore rewrite $A(s)$ in equation 12 as $\tilde{A}(\{(\lambda_{jf}(s), \theta_{jf}(s)\}_{j,f}) = \bigoplus_{j,f} \tilde{A}(\lambda_{jf}(s), \theta_{jf}(s))$, where $\tilde{A}_{jf} : \mathbb{R}^{\tilde{J}} \times \mathbb{T}^{\tilde{J}} \to \mathrm{gl}(V)$ is defined as[6]

$$\tilde{A}_{jf}(\lambda_{jf}, \theta_{jf}) = \begin{pmatrix} \lambda_{jf} & -r_f \theta_{jf} \\ r_f \theta_{jf} & \lambda_{jf} \end{pmatrix} .$$

This way, $\tilde{M}(\mathfrak{a}(s))$ is written as

$$\tilde{M}(\mathfrak{a}(s)) = \tilde{A}(\{(\lambda_{jf}(s), \theta_{jf}(s)\}) = \bigoplus_{j,f} \tilde{A}_{jf}(\lambda_{jf}(s), \theta_{jf}(s)) \tag{15}$$

$$= \bigoplus_{j,f} \begin{pmatrix} \lambda_{jf}(s) & -r_f \theta_{jf}(s) \\ r_f \theta_{jf}(s) & \lambda_{jf}(s) \end{pmatrix} \tag{16}$$

Thus, again with the abuse of notation, we practically define $\mathrm{Ctrl}_{\mathcal{E}}$ to be

$$\mathrm{Ctrl}_{\mathcal{E}}^{\mathrm{prac}} : x[0, s) \times a[0, s) \mapsto \{(\lambda_{jf}(s), \theta_{jf}(s)\}_{j,f} \subset (\mathbb{R} \times S_1)^{\tilde{J}, F}. \tag{17}$$

---

[5]We note that, in this section, we are omitting the additional index $n$ of slot attention architecture in order to avoid the over-complication of notations.

[6]Here, $\{r_f\}$ in implementation was chosen in a way similar to RoPE. Please see the attached code for the detailed setting.

where $S_1$ is the circle group. This practial Controller map $\mathrm{Ctrl}_{\mathcal{E}}^{\mathrm{prac}}$ maps the input through $\tilde{M}$, so the following model emerges as the actual modeling of the time series in CIP

$$x(t) = \Psi\left(\exp\left(\int_0^t \tilde{A}\left(\mathrm{Ctrl}_{\mathcal{E}}^{\mathrm{prac}}(x[0,s], a[0,s]))\right) ds\right) \Phi(x(0))\right). \tag{18}$$

We also remark that our $J$ in the main part of the manuscript corresponds to $J = \tilde{J} F$.

### A.5.1 REGARDING THE TIME DISCRETIZATION

Although our framework is built for continuous and compositional dynamics, a real-world dataset is bound to be temporally discrete, and $x \in C(I, \mathcal{X})$ is represented as a discrete sequence $\bar{x} = [x(\tau_m); m \in \mathbb{Z}]$ with $\tau_m < \tau_{m+1}$, and we will be forced to learn $\mathrm{Ctrl}_{\mathcal{E}}$ as well as $\Phi, \Psi$ with the piecewise interpolated model

$$A(s) = \sum_m A(\tau_m) 1_{s \in [\tau_m, \tau_{m+1})} := \sum_m \bar{A}[m] 1_{s \in [\tau_m, \tau_{m+1})}$$

where $\bar{A}[m] := A(\tau_m) = \tilde{A}(\{\lambda_{jf}(s), \theta_{jf}(s)\})$ is constructed from $(\lambda_{jf}(s), \theta_{jf}(s))$ defined analogously with the discretized $(\lambda_{jf}[m], \theta_{jf}[m])$. Putting $\Delta_m = \tau_{m+1} - \tau_m$, we therefore have

$$\int_0^{\tau_t} A(s) ds = \sum_{\ell < t} \bar{A}[\ell]\Delta_\ell$$

and the equation 13 becomes

$$x(t) = \Psi\left(\exp\left(\sum_{\ell < m} \bar{A}[\ell] \min(\Delta_\ell, \tau_\ell - t)\right) \Phi(x(0))\right) \tag{19}$$

Giving the same consideration to $\mathrm{Ctrl}_{\mathcal{E}}$, the predicted time series in CIP training defined with $\mathrm{Ctrl}_{\mathcal{E}}$ is therefore computed as

$$\hat{x}[m] = \Psi\left(\exp\left(\sum_{\ell < m} \tilde{A}\left(\mathrm{Ctrl}_{\mathcal{E}}^{\mathrm{prac}}(\bar{x}[:\ell], \bar{a}[:\ell])\Delta_\ell\right)\right) \Phi(x(0))\right) \tag{20}$$

$$= \Psi\left(\exp\left(\tilde{A}\left(\sum_{\ell < m} \left(\mathrm{Ctrl}_{\mathcal{E}}^{\mathrm{prac}}(\bar{x}[:\ell], \bar{a}[:\ell])\Delta_\ell\right)\right)\right) \Phi(x(0))\right) \tag{21}$$

where $\bar{a}$ is defined analogously to $\bar{x}$ and the notation of $\mathrm{Ctrl}_{\mathcal{E}}^{\mathrm{prac}}$ is abused to take as input the discretized version of $(x[0,s], a[0,s])$. In the final equality we used the linearity of $\tilde{A}$ with respect to its input.

### A.5.2 A REMARK REGARDING THE REPRESENTATION POWER OF OUR MODEL

Because our fundamental assumption is that all $G$ is a the group, every $g \in G$ is assumed invertible. Also, as we have mentioned earlier, we assume $G$ to be abelian as well. Therefore, strictly speaking, transitions operators that are nilpotent or idempotent cannot be expressed. While this might seem to restrict our modeling capability, our model can construct a transition that mimics these operators. Firstly, because $M(g(t)) = \bigoplus_k \exp(\Lambda(t)) \begin{pmatrix} \cos\Theta_k(t) & -\sin\Theta_k(t) \\ \sin\Theta_k(t) & \cos\Theta_k(t) \end{pmatrix}$ with $\Lambda_k(t) = \int_0^t \lambda_k(s)ds, \Theta_k(t) = \int_0^t \theta_k(s)ds$, we can create a sequence $M(g(t)) \to 0$ by considering $\Lambda(t) \to -\infty$. Likewise, because $\lim_{t \to t_*} \mathfrak{a}(t) \to 0$ for some fixed $t_* \in I$ is plausible for equation 12, we can also model a sequence similar to $g(t) \to g_*$ for some fixed $g_* \in G$ as well. In fact, in experiment, we are succeeding in modeling a complex dynamics in both real world setting like 1X World model dataset and simulated world setting like ProcGen which are both likely to involve transitions like nilpotent and idempotent operations.

## B   IMPLEMENTATION DETAILS

### B.1   PHYRE

The encoder was configured with a depth of 9 layers, a width of 216, a patch size of 8, 12 attention heads, and 6 slots. The decoder had a depth of 3 layers and a patch size of 16, converting the latent slots to images using the mechanism from slot attention (Locatello et al., 2020). For the attention modules on the patch tokens, we employed GTA (Miyato et al., 2023).

We trained the model using the AdamW optimizer with a learning rate of $5 \times 10^{-4}$ and a weight decay of 0.1 for 100 epochs. We set the sparsity regularizer scale $\alpha$ to 0.01. Linear warmup cosine annealing was applied with 10 warmup epochs and an initial learning rate of $1 \times 10^{-5}$.

### B.2   PROCGEN

For the encoder and the decoder, we used the same setting as for the Phyre experiment, except that the number of slots is 20. The IDM module, which converts latent states $z$ into latent action parameters $(\lambda, \theta)$, was implemented using a 1D convolutional architecture with SiLU activation and LayerNorm. We trained the model using the AdamW optimizer with a learning rate of $5 \times 10^{-4}$ over 200 epochs. We trained the model with 16 frames: 12 frames for prediction, followed by 4 burn-in frames.

When training $\text{Ctrl}_{adapt}$ with action labels (as described in Section 4.3), we input the action label sequences $a(t)$ together with the latent states $z(t)$ into a spatio-temporal transformer (Xu et al., 2020) configured with a depth of 4 layers and 4 attention heads. $\text{Ctrl}_{adapt}$ was trained using the AdamW optimizer with a learning rate of $5 \times 10^{-4}$ for 100 epochs. The encoder-decoder pair $(\Phi, \Psi)$ was frozen during the training of $\text{Ctrl}_{adapt}$.

### B.3   ANDROID DATASET

In this experiment, each $256 \times 256$ RGB video frame was converted into an $8 \times 8$ grid of 18-bit binary tokens using MAGVIT2. Consequently, we trained our model on the time series of these tokens and replaced the reconstruction losses with logistic losses, as the tokens are binary. Additionally, since Slot Attention expects images as input, we employed a standard Vision Transformer (ViT) architecture to model the time evolution and alignment of the encoded tokens. Specifically, since there are $8 \times 8 = 64$ tokens in total, we treat them as if they are "slot" tokens. Also, we set the number of rotation angles to $J = 16$.

### B.4   HYPERPARAMETER SELECTION

The number of slots $N$ and the number of Lie group actions $J$ play an important role. Generally, increasing $N$ and $J$ enhances the model capacity but also affects computational complexity, especially $N$, which directly impacts runtime. The same applies to $J$ since $D = 2J$.

We observed that the internal construction of $D$ significantly improves performance. Inspired by sinusoidal positional embeddings and NFT, we parameterized each rotation with a "frequency." We indexed $\theta$ with $i, j$ such that $\theta_{i,j}(t) = m_i \tilde{\theta}_j(t)$, where $\{m_i \in \mathbb{R} \mid i = 1, \ldots, F\}$ are the rotation speeds and $j \in 1, \ldots, \tilde{J}$ with $\tilde{J} = J/F$. We fixed $F$, with $D = 2J = \tilde{J}F$.

We conducted additional small-scale experiments on ProcGen to study the effect of varying $\tilde{J}$ and $N$. The results are summarized in the following table:

These results indicate that increasing $\tilde{J}$ with a smaller $F$ improves performance. Similarly, increasing $N$ reduces reconstruction error, highlighting the importance of these hyperparameters.

| $N$ | $\tilde{J}$ | Reconstruction Error (MSE) |
|---|---|---|
| 16 | 3 | 0.0583 |
| 16 | 6 | 0.0587 |
| 16 | 12 | 0.0439 |
| 16 | 24 | 0.0378 |
| 8 | 6 | 0.0641 |
| 16 | 6 | 0.0587 |
| 24 | 6 | 0.0397 |

Table 4: Effect of varying $\tilde{J}$ and $N$ on reconstruction error.

