# OpenReview forum: "Inter-Environmental World Modeling for Continuous and Compositional Dynamics"
_ICLR.cc/2025/Conference — Submitted to ICLR 2025_

### Official Review · Reviewer_em9c · 2024-10-27

**Soundness:** 2
**Presentation:** 2
**Contribution:** 2
**Rating:** 5
**Confidence:** 5

**Summary:**

The paper introduces World Modeling through Lie Action (WLA), a generative framework that learns continuous, compositional actions for controlling agents across diverse environments using Lie group theory. Claimed contributions include:

Unified Simulator: A shared, environment-agnostic simulator leverages continuous actions, unlike models with discrete, environment-specific setups.
Controller Interface Problem (CIP): WLA solves CIP with adaptable controllers, even under minimal labeling.
Object-centric Modeling: Slot-based object-centric modeling supports interactions across multiple objects and actions.
Performance: WLA seems to outperform on benchmarks (Phyre and ProcGen), proving its generalizability and adaptability.
This approach enhances state-space modeling with flexible, multi-environment simulations

**Strengths:**

The originality seems novel. The universal interface for control is an important problem to study. The emprical evidence seems to be strong, it can outperform the baseline significantly. I think there could be some contributions in the paper, but the presentation needs to be improved a lot.

**Weaknesses:**

The paper’s ambitious framework for inter-environmental modeling has notable weaknesses, especially in mathematical clarity and formulation, which hinder its accessibility and rigor.
 Key Issues:
1. **Ambiguity in Transition Operator (\( g_{t, \delta} \))**: \( g_{t, \delta} \) maps specific observations \( x(t) \) to \( x(t+\delta) \), making it appear as a trivial one point to one point mapping. This definition does not capture the desired dynamic evolution. It does not make any sense in the form written in the paper. The paper acknowledges that \( g_{t, \delta} \) depends on individual trajectories but simplifies it as a generic operator, risking confusion. This omission detracts from the model’s mathematical precision, especially for multi-environment dynamics.

3. **Underdefined Action Space \( A \)**: The action space \( A \) and action \( a(t, \delta) \) lack clarity on whether actions are fixed or variable over intervals. This ambiguity in structure reduces the framework's comprehensibility in continuous control scenarios.

4. **Non-Compositional Transition Operators**: Due to the triviality of \( g_{t, \delta} \), composing transitions over time is not feasible, so why does it form a Lie group? The paper introduces Lie groups without a clear justification for their relevance to the specific environments. Additional reasoning would strengthen its argument for using Lie groups to model compositional and continuous dynamics.

To improve clarity, the paper would benefit from revisiting the definitions and dependencies in its transition operators, better defining the action space, and providing justification for its mathematical choices. More precise formulation and notation would significantly enhance its accessibility and application.

**Questions:**

Revise the math definitions and formulations significantly and I will reconsider my score.

See weakness for all the math confusions.

---

> ### Author Response · Authors · 2024-11-15
> **Thank you for the Feedbacks (Part I)**
>
> We are sorry that we forwent the details of the mathematical setups.  Our formal mathematical settings are mostly built on the theory of NFT (Koyama et al., 2024), except that we use continuous and nonstationary time series for training the model.  We provide more details in our response below.   In the revision, we will prepare an Appendix section with more rigorous notations and mathematical setups.
>
> Please see the Part III for the summary of our responses to each component of the raised concerns.
>
>
> ## Regarding the assumption underlying WLA
> This section is meant to serve as our response to Q1 (Ambiguity in Transition Operator  $g_{t, \delta}$) and Q3 (Non-Compositional Transition Operators). To resolve the confusion, we will elaborate the assumption underlying WLA below with more formality.
>
>
> Let $D = \lbrace x^i: [0, T] \to \mathcal{X} \rbrace_i$ be the video dataset to be used to train the encoder $\Phi$ and the decoder $\Psi$ in our WLA framework, where $i$ is the sample index.  This dataset contains the videos sampled from multiple environments so $\mathcal{X}$ is a superset of observation space of all environments.
> The underlying assumption of WLA is that,  for every $i$, $t >0$ and $\delta>0$,  the transition $x^i(t) \to x^i(t+\delta)$ is *realized* by *some* common group action $G \times \mathcal{X} \to \mathcal{X}$ of a *some*  Lie Group $G$.
> To be more precise, our standing assumption is as follows:
>
> >**Assumption:** For our dataset $D= \lbrace x^i: [0, T] \to \mathcal{X} \rbrace_i$
> >consisting of movie samples from multiple environments, there exists some Lie group $G$ that acts on $\mathcal{X}$ through a group action $\alpha : G \times \mathcal{X} \to \mathcal{X}$
> >such that,  for every $x^i \in D$, $t >0$ and $\delta>0$ there is some $g^i_{t , \delta} \in G$ such that $\alpha ( g^i_{t , \delta}, x^i(t) )  = x^i(t+\delta)$.
>
> For brevity, we notationally identified $g$ with the map $\alpha_g: x \mapsto \alpha(g, x)$ ( that is, $g^i_{t , \delta}:  \mathcal{X} \to \mathcal{X}$  with  $g^i_{t , \delta}(x^i(t)) = x^i(t+\delta)$. ) We are sorry that the confusion might have been incurred by our typo of "$\to$" in place of "$\mapsto$".  We shall more correctly write $g^i_{t , \delta}: \mathcal{X} \to \mathcal{X}$.  We will fix this typo in the revision.
> With this assumption, we are therefore assuming that the family of transitions $T_\mathcal{E}$ can be identified with a subset of $G$ for every $\mathcal{E}$.
>
> We introduce this assumption of the Lie group structure as our inductive bias because our desiderata of compositionality and continuity matches (almost exactly, except for the invertibility) with the definition of Lie group as a topological set with smooth manifold structure and rules of compositions.  Note that, by leaving the group action and the details of $G$ unknown, we leave much freedom in our construction of the world model.
> We do note however that we assume $G$ to be abelian in order to make the latent transitions computationally simple.
>
>
> According to the theory and results presented in NFT, when there is some group $G$ acting on $\mathcal{X}$ with a set of reasonable regularity conditions,  there exists an autoencoder $(\Phi, \Psi)$ with latent vector space $V$ and a homomorphic map $M: G \to Aut_{\mathcal{R}}(V)$ such that $\Phi ( M(g) \Psi (x)) =  g(x)$.   Mitchel et al. (2024) and Miyato et al. (2023) had experimentally shown that such  $(\Phi, \Psi)$ and $M$ can be learned from a time series dataset, and our work scales their philosophy to the nonstationary, continuous dynamical system on large observation space.
> In our manuscript, this $M: G \to Aut_{\mathcal{R}}(V)$  derived from $\Phi$ and $\Psi$
> is written as $F_{\Phi, \Psi}:   G \to Aut_{\mathcal{R}}(V)$.
> As for the expression around eq (3) in concern, the continuity part of our statement more precisely reads as follows:
>
> >**Continuity:**
> >When each continuous transition $x^i(t) \to x^i(t + \delta)$ is realized by a group element $g^i_{t , \delta}$ so that
> >$\Phi ( M(g_{t, \delta}) \Psi (x^i(t))) =  x^i(t + \delta)$, then $\lim_{\delta \to 0} M(g_{t, \delta}) = \lim_{\delta \to 0} F_{\Phi, \Psi}(g_{t, \delta})  \to I$.
>
> In the revision we will strike out ambiguous expression like "when the set of transitions form a Lie group".
> We chose the modeling of the transitions like (Koyama et al, Miyato et al and Mitchel et al) because (1) if $\Phi$ and $\Psi$ are continuous and invertible, the compositionality and continuity of $M$ directly lift to the those of $G$'s action on $\mathcal{X}$, and because (2) $M(g)$ for each $g$ can be represented by sparse (block diagonal) matrix that is computation-friendly (assuming that finite-dimensional representation space with sufficiently large dimension can reproduce the observations).    That is, we use $F_{\Phi, \Psi}$ as the environment-agnostic encryptor of the underlying transition operators.

---

> ### Author Response · Authors · 2024-11-15
> **Thank you for the feedbacks (Part II )**
>
> ## Regarding the Action space $\mathcal{A}$
> This section is meant to serve as our response to Q2 (Underdefined Action Space). To resolve the confusion, we elaborate our the defintiion and the role of $A$ in WLA with more formality.
>
> We intentionally left the action space $\mathcal{A}$ as abstract as possible, because it is meant to represent the set of "controller inputs" of an unknown interface.  At the minimum, let us assume that $\mathcal{A}$ is a topological space. $\mathcal{A}$ may depend on $\mathcal{E}$, so that we shall more precisely write $A_E$ (please forgive us in writing this as $A_E$ in this response, because there is a problem with Markdown.)
> With this in mind, let us elaborate on where $A_{E}$ stands in the WLA framework.
>
> Our general assumption is that the time series in the world model can be controlled with continuous time series of action inputs.
> More precisely, let $I=[0,T]$ be the time span and $C(I, A_E ) \subset \lbrace a |   I \to A_E \rbrace $ be the space of the control sequences. In an application, we are literally expecting each element of $C(I, A_E )$ to be the sequence of inputs from the controller interface.
> We assume that there exists a controller interface map
>
> $$Ctrl_E :   C(I, A_E) \times  C(I, \mathcal{X}) \times I \to  T_{id}G$$
>
> where $T_{id}G$ is the tangent space of $G$ at $id$, or the Lie Algebra of $G$. Because $Ctrl_E$ is "a controller" whose effect depends only on the past,  we assume that $Ctrl_E(a, x, t)$ depends only on $a[0, t) = \lbrace a(s) |  s \in [0, t) \rbrace$ and $x[0, s) = \lbrace x(s) |  s \in [0, t) \rbrace$. Therefore let us  write  $Ctrl_E(a, x, t) :=  Ctrl_E(a[0, s), x[0, s))$.
>
> It is for brevity we wrote $Ctrl_E : A_E \times \mathcal{X} \to T_E$ in the manuscript to convey the general idea that the controller input is mapped to transitions, but we are sorry if this caused confusion.  In the revised Appendix section, we will add more formal explanations as we presented here.
>
> In the training of  $Ctrl_E$,  we are assuming a "labeled" sequential dataset of small size
> $D_{label} = \lbrace (a_i, x^i)  \rbrace_i \subset C(I, A_E \times \mathcal{X}) $
> such that, with $(\Phi, \Psi)$ and $M$ (the learned representation of $G$)  in place,
>
> $$x^i(t) = \Psi \left( M \left( \exp \left(  \int_0^t Ctrl_E(a_i[0, s), x^i[0, s) ) ds  \right) \right) \Phi (x^i(0)) \right). $$
> The purpose of Controller Interface Problem is to learn $Ctrl_E$ from $D_{label}$, as described in the manuscript.
> Note that, in this expression, we intend
> $\exp \left(  \int_t^{t+\delta} Ctrl_E(a_i[0, s) , x^i[0,s)) ds  \right)$  to represent $g^i_{t, \delta}$ in the first part of our response, so that
> $g^i_{t, \delta}$ depends on $ \lbrace Ctrl_E(a_i[0, s), x^i[0,s) | s \in [t, t+\delta) \rbrace.  $

---

> ### Author Response · Authors · 2024-11-15
> **Thank you for the feedbacks (Part III-1,  answers to  specific components of the raised concern)**
>
> We make sure that we make revisions according to our answers above.
> That being said,  our specific responses to each components of the three Questions are as follows.
>
> **Q1 (Ambiguity in Transition Operator)**
>
> > ( $g_{t, \delta}$ ) maps specific observations ( $x(t)$ ) to ( $x(t+\delta)$ ), making it appear as a trivial one point to one point mapping. This definition does not capture the desired dynamic evolution. It does not make any sense in the form written in the paper.
>
> We are sorry for our typographical use of $\to$ in place of $\mapsto$.  We assume that all transitions in all environments are the dataset are caused by actions of Lie group $G$, and we meant $g_{t, \delta, i}: \mathcal{X} \to \mathcal{X}$ to indicate a group action of 'an' element $g_{t, \delta, i}$ such that $g_{t, \delta, i}(x^i(t)) :=g_{t, \delta, i} \cdot x^i(t) = x^i(t+ \delta)$.
>
> > The paper acknowledges that ( $g_{t, \delta}$ ) depends on individual trajectories but simplifies it as a generic operator, risking confusion.
>
> Indeed,  at the time of the training, $g_{t, \delta} \in G $ should be more correctly labeled as $ g_{t, \delta, i}$ to represent a group that is assumed to realize $g_{t, \delta, i}(x^i(t))= x^i(t+ \delta)$.  It was our wish to omit $i$ when we want to discuss a generic random trajectory $x$.  In the revision, we make sure to use $i$ when the label is important.
>
> > This omission detracts from the model’s mathematical precision, especially for multi-environment dynamics.
>
> Because our underlying assumption is that there is some "common" $G$ that realizes $\mathcal{T}_\mathcal{E}$ for all $\mathcal{E}$, we believe that the absence of $i$  in the notation is orthogonal to the discussion of multi-environment dynamics.
>
> **Q2 (Underdefined Action Space)**
>
> > The action space ( $A$ ) and action ( $a(t, \delta)$ ) lack clarity on whether actions are fixed or variable over intervals. This ambiguity in structure reduces the framework's comprehensibility in continuous control scenarios.
>
> We are sorry that the informal notation of $a(t, \delta)$ might have caused confusion.
> More formally, we assume that $A$ is some topological space, and  our underlying assumption in CIP is that there exists some ground-truth map that maps a _time series_ $a : I \to A$ to a _time series_ on $G$ (e.g  $g : I \to  G$).
>
> Because we assume $a \in C(I, A)$, where $C(I, A)$ is the set of all continuous map from the time span $I \subset R_+$ to $A$,   our  $a(t)$ is assumed to change with time $t$.  However, because the implementation itself must be conducted discretely, $a(t)$ is assumed to be fixed over $[t_i, t_{i+1}]$ with $t_{i+1}- t_i = \delta$ and $t_i = t$, and we originally meant $a(t, \delta)$ to informally represent $\lbrace a(s) | s \in [t, t+\delta) \rbrace = \lbrace a(s) | s \in [t_i, t_{i+1}) \rbrace $ that takes the same value $a(t)$ over the interval $[t, t+\delta]$.  We admit that this can be a confusing notation, so we will not use the notation of $a(t, \delta)$ in the revision and instead add more comments to our description regarding discretization (Below eq (5)).

---

> ### Author Response · Authors · 2024-11-15
> **Thank you for the feedbacks (Part III-2, answers to specific components of the raised concern)**
>
> **Q3 (Noncompositionality of $g_{t, \delta}$)**
>
> > Due to the triviality of ( $g_{t, \delta}$ ), composing transitions over time is not feasible, so why does it form a Lie group?
>
> Once again, our assumption is that all transitions $\mathcal{T}_E$ in the video dataset are realized by some common group action of some Lie group, but $\mathcal{T}_E$ is not necessarily the Lie group itself.  We agree that our expression of "transitions forming a Lie group" can be misleading, so we will strike such an expression. Our intension regarding the composability is referring to the composability of transition *operators* (action of group elements), and they can indeed be composed because we are assuming them to be a member of a Lie group.
>
> It is both theoretically and experimentally verified in the past works (Koyama et al, Mitchel et al, Miyato et al) that the representations of hidden group actions can be found through the autoencoder in which the the transisions are modeled linearly in latent space, and it has been verified that the learned group actions are indeed composable.  The learning of $\Phi, \Psi$ in WLA scales their works to non-stationary video dataset of high complexity.
>
> > The paper introduces Lie groups without a clear justification for their relevance to the specific environments. Additional reasoning would strengthen its argument for using Lie groups to model compositional and continuous dynamics.
>
> We would like to emphasize that our goal is not to learn an interactive model for a specific environment. Rather, we aim to learn a universal model that can be generalized across multiple environments with continuous dynamics.
> As we stated above, we assume that all transitions in all environments are realized by a common Lie group. We use the Lie group because our goal is (1) to uncover a family of environment-agnostic transition operators that are composable and differentiable with respect to time, and (2) to use the family to learn an interface that can be used to control various environments.
>
> Lie group is almost a forced choice in modeling such a family of transition operators because a _topological set that is both smooth (differentiable) and is closed under a consistent rule of composition_ is the almost definition of Lie group (a group is a *set* with identity that is closed under composition and inversion).  We would also like to add that because the representations of Lie group can be expressed as direct sum of matrices, and this benefits us computationally in composing the transition operators both at the time of training and inference (especially when we use abelian Lie group. )

---

> > ### Author Response · Authors · 2024-11-21
> > **A gentle reminder regarding the openreview dicussion of "Inter-Environmental World Modeling for Continuous and Compositional Dynamics"**
> >
> > Dear Reviewer em9c,
> >
> > We hope this message finds you well. We wanted to kindly remind you that we have provided detailed responses to your feedback on our paper, "Inter-Environmental World Modeling for Continuous and Compositional Dynamics." We greatly appreciate the time and effort you've invested in reviewing our work.
> >
> > Your insights are invaluable to us, and we are committed to addressing all your concerns thoroughly. We would be grateful if you could consider our responses and let us know if there are any further questions or clarifications we can provide.
> >
> > Thank you once again for your thoughtful review.
> >
> > Best regards,
> > The Authors

---

> > > ### Comment · Reviewer_em9c · 2024-11-26
> > >
> > > I am raising my point to 5, after the author's clarification.
> > >
> > > I believe there are still some problems. For example, the author assume that these actions are always reversible, in most cases this is not true. I think there are still some potential problems, but the expressions are much better now.

---

> ### Author Response · Authors · 2024-11-27
> **Thank you for the followup!**
>
> Thank you for reconsidering your score and for your continued engagement with our work. We understand that you still have some concerns, particularly regarding our assumption of reversibility of actions. We would like to address them.
>
> > For example, the authors assume that these actions are always reversible; in most cases, this is not true.
>
> We appreciate your feedback and would like to clarify our assumptions in this regard. In our framework, it is important to make a distinction between the following two types of meaning for the word “action”, dealt separately in the training of $(\Phi, \Psi)$ and in CIP:
>
> 1. **The group action $g$ and the transition of data $x$ modeled by $g$.**
> 2. **The external action signal $a$.**
>
> Since it was unclear to us which one your question refers to, we will address both.
>
> **1. Reversibility of the Group Action $g$ and Data Transitions**
>
> It is true that the group action $g$ in our model is invertible (reversible), because in group theory, every element has an inverse.Moreover, we consider deterministic processes, where the forward transition from $x(t)$ to $x(t+\delta)$ is one-to-one. This means the inverse mapping from $x(t+\delta)$ back to $x(t)$ is always well-defined [1].
>
> While this assumption in our modeling may seem restrictive, note that “every video can be *played* backward” in real world applications. 　For example, in the real physical world, we cannot directly “undo” the operation of  throwing a water balloon against the wall. However, we can surely “imagine” the undoing process of gathering the scattered water drops and repairing the balloon by enacting imaginary forces. In modeling, our group action of $G$ contains such an “imaginary” operators as well, but this is not to be confused with those triggered by the external action signal we describe below.
>
>
> **2. Reversibility of the External Action Signal $a$**
>
> When we solve the Controller Interface Problem (CIP), we create a mapping from the external action signal $a$ and observation $x$ to $g$ via the Controller **Ctlr**. For example, suppose we are playing a Mario game. If Mario falls from a cliff higher than he can jump, no matter what button operations we perform (unless we restart the stage), we cannot return to the state before he fell off the cliff.
>
> Let us consider the set of operators that the controller can output when the player is in state $x$, denoted as $\mathcal{T}[x] \subset G$. When $x$ is the state of Mario at the bottom of the cliff, the example above illustrates the fact that  $\mathcal{T}[x]$ does not necessarily include elements that can reverse the action of "falling off the cliff." Fortunately, such situations are within the scope of our controller's design. This is because our controller is designed to take the history of $a$ and $x$ as input, so $\mathcal{T}[x]$ depends not only on $x$ but also on the history of $x$.
>
> As an extreme case, if we have $\mathcal{T}[x] = G$, the model would simulate operations as if it is in a debug mode. This is not what we usually want to do, however,  because we aim to obtain a controllable simulator that is faithful to the actual dynamics. We can avoid this scenario by property training the controller **Ctlr**.
>
> ---
>
> In summary, we would like to emphasize that the reversibility of $a$ and the reversibility of $g$ are distinct considerations. The transition $g$ results from both $a$ and $x$, and our assumption of reversibility in $g$ does not constrain our theory. Moreover, we would like to point out that the reversible assumption of G doesn’t hurt the model’s performance. Our experiments show WLA outperforms Genie numerically in both 2D and 3D environments.
>
> If you have additional concerns about the reversibility of actions or other aspects of our work, we would be grateful if you could specify them. We are committed to addressing all your concerns thoroughly.
>
>
> [1] [https://en.wikipedia.org/wiki/Time_reversibility](https://en.wikipedia.org/wiki/Time_reversibility)

---

### Official Review · Reviewer_1yM7 · 2024-11-02

**Soundness:** 2
**Presentation:** 4
**Contribution:** 3
**Rating:** 6
**Confidence:** 2

**Summary:**

In this paper, the authors present World modeling through Lie Action (WLA), a generative state-space-modeling framework that can be trained in an unsupervised fashion on interactive environments, and that allows controllable generation of future frames of a given environment. One main goal of this framework is learning continuous and compositional action representations, similar to what humans are able to do. It is a step towards an interactive world model that generalizes across environments, given these environments have a common basic rules of composition and continuity.
The paper presents a model built using this framework that is benchmarked on two datasets (ProcGen and Phyre). In general, the model can be trained using no or only few action labels, making it more versatile and faster to adapt to new settings with different action labels.

**Strengths:**

1. The paper is clearly structured, and the individual components of the model with accompanying formulas, as well as the setting is well introduced. The authors carefully set the scope of the work and how it compares to similar approaches.
2. Benchmarks with different test settings (e.g. different FPS compared to train) shows the robustness of the model on this benchmark and the ability to infer continuous dynamics.
3. Given a main focus of the framework is the Lie group theory, I appreciate the ablation study that reveals the relevance of it (Rotation) to the predictive performance of the model

**Weaknesses:**

My main concern is the limited amount of comparability to previous works and approaches. The model is only benchmarked on two datasets, and one is mainly used as a (successful) sanity check. The second dataset (with 16 quite diverse environments) is benchmarked against only one other model, though using several metrics. Even if it is a ‘first of its kind’ framework as stated by the authors, more benchmarks to confirm its proper function would increase trust in the method/framework. It can be acknowledged though that the number of available datasets/benchmarks for this specific scope is low, as also stated in a recent survey [1].
Other things that are not clear to me at the moment are in the question section.


[1] McCarthy, Robert, et al. "Towards Generalist Robot Learning from Internet Video: A Survey." arXiv preprint arXiv:2404.19664 (2024).

**Questions:**

1. Was the encoder and decoder trained from scratch?
2. Did you do multiple runs with different random initializations, or are the results from just one experimental run?

---

> ### Author Response · Authors · 2024-11-20
> **Thank you for your feedbacks**
>
> Thank you for your positive feedback and thoughtful review. We appreciate your insights and address your concerns and questions below.
>
> ## **Response to Weaknesses**
>
> ### **W1. Limited Experiments**
>
> > *My main concern is the limited amount of comparability to previous works and approaches. The model is only benchmarked on two datasets, and one is mainly used as a (successful) sanity check.*
>
> To address this concern, we have conducted an additional experiment on a large video dataset of robot actions in real-world environments (the 1X World Model dataset). This dataset contains over 100 hours of recordings of android-type robots manipulating objects in various settings, including narrow hallways, spacious workbenches, and tabletops with multiple objects under different lighting conditions. We compared our method against Genie using various metrics. While our method slightly underperforms Genie in terms of per-frame fidelity (PSNR), it significantly outperforms Genie on the temporally local fidelity ($\Delta t$ PSNR) and the video-specific metric (FVD).
>
> To further support these quantitative measurements, we will include video demonstrations of our world model control in the supplementary material. In these videos, comparing our method against ground truth and Genie, we observe that our method predicts much more contextually realistic responses to the control sequences.
>
> For more details, please refer to our general response G1.
>
> ---
>
> ## **Response to Questions**
>
> ### **Q1. Was the encoder and decoder trained from scratch?**
>
> Yes, both the encoder and decoder were trained from scratch without using any pretrained models from other datasets.
>
> ### **Q2. Did you do multiple runs with different random initializations, or are the results from just one experimental run?**
>
> Due to limited computational resources, we report results from single runs.

---

> > ### Comment · Reviewer_1yM7 · 2024-11-25
> >
> > Thank you to the authors for providing these additional details and considering the inputs.

---

> > > ### Author Response · Authors · 2024-11-28
> > >
> > > Thank you for your response. Discussions with the other reviewers are still progressing, and their concerns are being resolved based on the updated revision and the added experimental results.
> > >
> > > If you have any remaining concerns regarding the currently updated version or the ongoing discussion,  we are committed to addressing them. Please let us know.
> > >
> > > Thank you for your time and consideration.

---

### Official Review · Reviewer_vJwu · 2024-11-04

**Soundness:** 3
**Presentation:** 2
**Contribution:** 3
**Rating:** 6
**Confidence:** 3

**Summary:**

The paper proposes a method for enforcing a Lie group structure on the latent space of a multi-environment world model.  This structure imposes compositionality and continuity in the latent space, and allows the dynamics in the latent space to be linear.  The latent space is further structured by imposing object-specific dynamics (through the use of slot attention and assignment of "close" slots in nearby time steps).  This method allows to learn a common dynamic model from a sequence of observations sampled from a set of environments, and the paper shows that this model can generate observations that better reconstruct the ground truth observation compared to a strong baseline.

**Strengths:**

- The paper presents an interesting idea, and a clever method for exploiting similarities between several environments at once.
- Learning reliable world models from observation trajectories without paired actions would be a dramatic step forward, and this paper presents a novel angle on that problem.
- Evaluation shows improvements over a strong baseline.

**Weaknesses:**

- In places the presentation can be a bit unclear.  This is mostly in the description of the method.  I would have benefitted from a diagram of the steps required to train WLA and then use that to solve the CIP.  The current Figure 1 may not be necessary; the inter-environment aspect of WLA is probably the most straightforward step in the process.
- Although not direct alternatives to the latent space presented here, there is some previous work in structured latent spaces that it may be interesting to compare against in the related work section.  eg.
    - Embed to Control: A Locally Linear Latent Dynamics Model for Control from Raw Images.  Manuel Watter, Jost Tobias Springenberg, Joschka Boedecker, Martin Riedmiller
    - Object Files and Schemata: Factorizing Declarative and Procedural Knowledge in Dynamical Systems.  Anirudh Goyal, Alex Lamb, Phanideep Gampa, Philippe Beaudoin, Sergey Levine, Charles Blundell, Yoshua Bengio, Michael Mozer
- Not a weakness, but Line 111 contains a reference to OCCAM, which I presume is a previous name for WLA?

**Questions:**

- How are the parameters $N$ and $J$ chosen?  Do they need to scale with the complexity of the environment?  If so, how do we expect computation costs to scale?
- Line 192 suggests an assumption about infinitesimal changes to observations between timesteps.  This assumption seems very strong, how would the method handle larger changes?
- How similar do the environments need to be for the inter-environment simulator to model them jointly?  For example, could you mix the Phyre and ProcGen environments?
- Does it matter how the dataset is sampled from each environment?

---

> ### Author Response · Authors · 2024-11-20
> **Thank you for the Feedbacks (Part I)**
>
> Thank you for your constructive comments and feedback. We address each of your concerns and questions below.
>
> ---
>
> ## **Response to Weaknesses**
>
> ### **W1. Presentation**
>
> > *In places the presentation can be a bit unclear. This is mostly in the description of the method. I would have benefitted from a diagram of the steps required to train WLA and then use that to solve the CIP. The current Figure 1 may not be necessary; the inter-environment aspect of WLA is probably the most straightforward step in the process.*
>
> Thank you for highlighting this issue. We will revise our figures to include a detailed diagram that illustrates the training steps of WLA and how it is utilized to solve the CIP. We will also consider modifying or removing Figure 1 to improve the clarity and focus of our presentation.
>
> ### **W2. Related Work**
>
> > *Although not direct alternatives to the latent space presented here, there is some previous work in structured latent spaces that it may be interesting to compare against in the related work section.*
>
> We appreciate you bringing these relevant works to our attention. We will include discussions of "Embed to Control: A Locally Linear Latent Dynamics Model for Control from Raw Images" (Watter et al.) and "Object Files and Schemata: Factorizing Declarative and Procedural Knowledge in Dynamical Systems" (Goyal et al.) in the related work section.
>
> In particular, the SCOFF model from "Object Files and Schemata" shares similarity with our approach because our approach too introduce concepts analogous to schemata and object files. In our work, the irreducible components of the Lie group action (such as abstract rotation and scaling) function similarly to schemata, and we use slot attention mechanisms akin to object files. Our key difference lies in equipping these schemata with structures of continuity and compositionality, built upon a Lie group structure shared across multiple environments. This structure acts as an environment-agnostic dictionary of dynamics.
>
> Regarding "Embed to Control" (E2C), while it also advocates controlling dynamical systems in latent space, our work fundamentally differs from E2C in that we combine algebraic latent modeling with frameworks like VPT and Genie to separate the causal part of the model from the non-causal part. WLA learns the latent structure in the non-causal part without control sequences, capturing environment-agnostic symmetric structures of the observation space that is independent of causal mechanisms. This separation allows us to model environment-dependent causal dynamics and control inputs distinctly. E2C's ideas are complementary to ours, and applying our scheme to more application-specific tasks is an avenue for future work.
>
> ### **W3. Typo**
>
> > *Not a weakness, but Line 111 contains a reference to OCCAM, which I presume is a previous name for WLA?*
>
> Thank you for catching this typo. You are correct; "OCCAM" should be "WLA." We will correct this in the revision.

---

> > ### Author Response · Authors · 2024-11-20
> > **Thank you for the Feedbacks (Part II)**
> >
> > ## **Response to Questions**
> >
> > ### **Q1. On Hyperparameter Selection**
> >
> > > *How are the parameters $ N $ and $ J $ chosen? Do they need to scale with the complexity of the environment? If so, how do we expect computation costs to scale?*
> >
> > Thank you for this important question. The parameters $ N $ (the number of slots) and $ J $ (the number of Lie group actions) are crucial for controlling the model's capacity. Generally, larger values of $ N $ and $ J $ enable the model to capture more complex dynamics.
> >
> > To choose $ N $ and $ J $, we consider the complexity and diversity of the environments we are modeling. For more intricate environments, we increase $ N $ and $ J $ to ensure sufficient capacity. However, increasing these parameters impacts computational costs: the cost scales linearly with $ N $ and quadratically with $ J $. There is a trade-off between model capacity and computational efficiency, and we select $ N $ and $ J $ based on preliminary experiments that balance the performance and the resource constraints.
> >
> > We have comprehensively discussed the choice of $N$ and $J$ in our general response G2. We have also discussed the computational complexity in our response W1 to Reviewer N9Uv. Please refer to these responses for more details.
> >
> >
> > ### **Q2. Infinitesimal Transition**
> >
> > > *Line 192 suggests an assumption about infinitesimal changes to observations between timesteps. This assumption seems very strong; how would the method handle larger changes?*
> >
> > We apologize for any confusion caused by the term "infinitesimal transition." Our intention was to indicate that the time increments in the dataset are small enough (i.e., the time resolution is high) to capture continuous dynamics.
> >
> > In the discretely sampled time series $(x(t), x(t+1))$, we are effectively capturing $(x(t), x(t+\Delta t))$, where $\Delta t = T/N$ is the time step of the evolution. This leads to the approximation:
> >
> > $$
> > \exp \left( \int_0^T A(s) ds \right) \approx \exp \left( \sum_{n=1}^{N} A\left( \frac{n}{T} \right) \Delta t \right).
> > $$
> >
> > This does not restrict our model to slow-changing dynamics. Even if $\Delta t$ is small, the product $A(n/T) \Delta t$ can represent significant changes. For instance, in the sample movie `supp/gif/seen_in-play_ninja.gif` provided in the supplementary material, we capture rapid transitions like explosions.
> >
> > However, we acknowledge that if the dataset has very coarse time resolution, our model may struggle to interpolate the dynamics effectively.
> >
> > ### **Q3. Similarities of Environments**
> >
> > > *How similar do the environments need to be for the inter-environment simulator to model them jointly? For example, could you mix the Phyre and ProcGen environments?*
> >
> > Hypothetically, we could mix Phyre and ProcGen environments when training the encoder and decoder, as the transitions in Phyre model the 2D object movements similar to those in ProcGen. These dynamics shall belong to the same family, based on our intuitive assessment.
> >
> > However, training the environment-specific controller $ \text{Ctrl}_E $ for Phyre cannot be done in the same way as for ProcGen Dataset,  because Phyre does not provide a supervision of controlling signals. Beyond obtaining the trivial identity controller in an unsupervised manner (where $ A = G $ itself), we may not effectively model environment-specific causal dynamics for Phyre without additional supervision. The composition results shown in Figure 4 is realized by the trivial identity controller.
> >
> > ### **Q4. Sampling of Dataset**
> >
> > > *Does it matter how the dataset is sampled from each environment?*
> >
> > Our method does not rely on specific sampling strategies. As long as the trajectories are sampled uniformly at random from each environment, the training process should be successful. Uniform random sampling ensures that the model captures the diverse dynamics present in the environments.
> >
> > ---
> >
> > Once again, we appreciate your thoughtful feedback and hope our responses address your concerns. We will incorporate the suggested revisions to improve the clarity and completeness of our paper.

---

> ### Comment · Area_Chair_KE8n · 2024-11-26
> **[ACTION NEEDED] Respond to author rebuttal**
>
> Dear Reviewer,
>
> Now that the authors have posted their rebuttal, please take a moment and check whether your concerns were addressed. At your earliest convenience, please post a response and update your review, at a minimum acknowledging that you have read your rebuttal.
>
> Thank you,
> --Your AC

---

### Official Review · Reviewer_N9Uv · 2024-11-04

**Soundness:** 1
**Presentation:** 3
**Contribution:** 3
**Rating:** 5
**Confidence:** 3

**Summary:**

The paper proposes a method akin to state-space models for simulating environments by applying Neural Fourier Transform. It uses Lie group theory to obtain a continuous representation of the environment dynamics.

**Strengths:**

- The paper is clearly structured and well written
- The use of the Lie group theory and NFT to model the dynamics of interactive environments seems to be both novel and interesting
- The method is well motivated and theoretically grounded
- Good reproducibility (code provided)

**Weaknesses:**

- No runtime analysis
- No discussion of hyperparameter selection
- No significance tests
- No error bars
- No scaling laws
- No quantitative results on Phyre

Unfortunately, the empirical evaluation does not meet scientific standards and, therefore, I do not deem the paper ready for publication yet. Once these problems are solved, I think this paper will constitute a nice contribution to the field.

**Questions:**

- In Eq. (1), does the function Ctrl return an observation or a transition operator?
- In Eq. (3), shouldn't one take the limit of delta to 0 instead of t to infinity?
- Are there any limitations arising from the restriction of transitions to Lie groups?
- Since the latent space is divided into different slots for different objects and the slots forward dynamics are independent of each other, how can the model represent interactions between objects?
- It seems that the size of the latent space is a crucial hyperparameter; how was it determined (both N and J)?

---

> ### Author Response · Authors · 2024-11-20
> **Thank you for the Feedbacks (Part I)**
>
> Thank you for your thoughtful review and valuable comments. We have addressed your concerns below.
>
> ## **Response to Weaknesses**
>
> ### **W1. No runtime analysis**
>
> We appreciate your suggestion to include a runtime analysis. While recent studies in our field, such as Genie and VPT, do not provide detailed runtime analyses due to the architecture-agnostic nature of their frameworks, we recognize the importance of this information.
>
> We provide below the computational complexity of our framework when using a Transformer with single-head attention. Let us define:
>
> - $T$: Number of video frames
> - $N$: Number of slots
> - $D$: Embedding dimension
> - $P$: Number of image patches
>
> The computational complexities of each module in our framework are as follows:
>
> | Module                        | Computational Complexity                |
> |-------------------------------|-----------------------------------------|
> | Encoder $\Phi$                | $O(TNPD + TND^2)$                       |
> | Decoder $\Psi$                | $O(TNP^2 D + TNPD^2)$                   |
> | Time Evolution (applying $M$) | $O(TND)$                                |
> | Controller $Ctlr_{\text{adapt}}$ | $O(N^2 D + T^2 D + ND^2)$             |
>
> In the revision, we will include a section in the Appendix detailing these parameters.
>
> The two major sources of computational cost in training our encoder are:
>
> 1. **Cross-attention between slots and patches**: $O(TNPD)$
> 2. **Feed-forward network**: $O(TND^2)$
>
> The decoder computes self-attention for the patches of every slot, requiring $O(TNP^2 D)$. For time evolution, we compute the element-wise products for scaling and rotation, utilizing a RoPE-like computation trick since the matrix $M$ is diagonal. The controller $Ctlr_{\text{adapt}}$ is implemented as a spatio-temporal transformer, consisting of a stack of self-attention layers dedicated separately to time and slots.
>
> ### **W2. No discussion of hyperparameter selection**
>
> Thank you for highlighting the importance of hyperparameter selection. Please refer to our general response G2.
>
> ### **W3. No significance tests and no error bars**
>
> We acknowledge the importance of significance tests and error bars for enhancing transparency. However, in the domain of world models, there is no standard protocol for reporting statistical significance, and prominent studies like Genie do not provide such reports. Datasets like Phyre and ProcGen offer only two data splits (train/test), limiting the ability to perform significance tests.
>
> Training world models multiple times to compute statistical significance is also very computationally demanding. Nonetheless, we understand the value of such evaluations and have conducted an additional large-scale experiment on videos of robot actions in real-world environments. Please refer to our general response G1 for details on this experiment.
>
> ### **W4. No scaling laws**
>
> We agree that analyzing scaling laws can provide valuable insights. However, scaling laws are typically used to support claims about foundational models whose generalization abilities improve monotonically with increased data and model size, potentially learning abilities not directly accessible in the dataset.
>
> Our current work does not aim to present a foundational model but rather to introduce an algebraically inspired world-model framework of compositional and continuous actions that rivals existing schemes like Genie.
>
> Moreover, analyzing scaling laws requires computational power and datasets of immense scale, which far exceeds our computational budget. We believe that such an extensive study would warrant a separate research project.
>
> ### **W5. No quantitative results on Phyre**
>
> We used Phyre primarily for qualitative demonstrations, as it does not provide labeled action-control sequences required for the quantitative validation of CIP against other methods. Therefore, we did not include numerical results.
>
> To enhance the credibility of our framework, we have conducted an additional large-scale experiment on videos of robot actions in real-world environments. Please refer to our general response G1 for more information.

---

> > ### Author Response · Authors · 2024-11-20
> > **Thank you for the Feedbacks (Part II)**
> >
> > ## **Response to Questions**
> >
> > ### **Q1: In Eq. (1), does the function Ctrl return an observation or a transition operator?**
> >
> > Thank you for this question. In Eq. (1), the function $\text{Ctrl}$ returns a transition operator in the form of parameters for our Lie-transition model. Specifically, $\text{Ctrl}$ outputs parameters of a representation of the Lie algebra, whose exponential maps to the Lie group representation. In our implementation, $\text{Ctrl}$ returns $\exp[ A(\lambda, \theta) \Delta t ]$, where we set $\Delta t = 1$, assuming a high time resolution in the time series.
> >
> > ### **Q2: In Eq. (3), shouldn't one take the limit of $\delta$ to 0 instead of $t$ to infinity?**
> >
> > Thank you for pointing out this typo. You are correct; it should indeed be $\delta \to 0$.
> >
> > ### **Q3: Are there any limitations arising from the restriction of transitions to Lie groups?**
> >
> > Yes, restricting transitions to Lie groups imposes certain limitations. Our framework assumes that all transitions are invertible, which means it cannot directly model non-invertible transitions such as idempotent operations ($g^n = g^{n+m}$) or nilpotent operations ($M(g)^m = 0$ for some $m$). Additionally, because we use a commutative Lie group, the transitions satisfy $g_1 g_2 = g_2 g_1$.
> >
> > However, we can address these limitations in practice due to our non-autonomous time-series model:
> >
> > 1. **Approximation of Nilpotent and Idempotent Behaviors**: Our Lie group includes both scaling ($\lambda$) and rotations ($\theta$), allowing us to approximate nilpotent and idempotent behaviors when $\lim_{s \to t_*} \lambda(s) \to -\infty$ or $\lim_{s \to t_*} \lambda(s) \to 0$.
> >
> > 2. **State-Dependent Control**: By incorporating state-dependency in $\text{Ctrl}$, we can model behaviors like $g^n = g$ if $\text{Ctrl}$ learns to output an identity transition when the previous state is $g \circ x$ (e.g., preventing an action that has already been performed).
> >
> > Similarly, we can model non-commutative responses to input actions through a state-dependent controller.
> >
> > ### **Q4: How can the model represent interactions between objects?**
> >
> > Interactions between objects are managed by $Ctlr_{\text{adapt}}$. As mentioned in Line 465, $Ctlr_{\text{adapt}}$ is implemented using a spatio-temporal transformer that alternates between spatial and temporal attention. The spatial attention is applied to the slots, allowing interactions between them. In contrast, the encoder and decoder treat slots independently, and no interactions occur there.
> >
> > ### **Q5: The size of the latent space is a crucial hyperparameter; how was it determined (both $N$ and $J$)?**
> >
> > Indeed, the sizes of $N$ and $J$ are crucial hyperparameters. We determined them based on empirical observations and computational considerations. As detailed in our response to the hyperparameter selection concern (W2), increasing $N$ and $J$ generally improves performance but also increases computational complexity.
> >
> > We conducted experiments varying $N$ and $\tilde{J}$ to study their effects on the reconstruction error, as shown in the table provided earlier. We will include a comprehensive discussion of hyperparameter selection in both the main text and the Appendix.
> >
> > ---
> >
> > Thank you again for your constructive feedback. We are confident that addressing these concerns will enhance the quality of our paper, and we look forward to any further comments you may have.

---

> > > ### Comment · Reviewer_N9Uv · 2024-11-25
> > >
> > > Thank you for your answer. I appreciate the additional experiments and analyses. They address some (yet not all) of my concerns. I will adjust my score accordingly.
> > >
> > > Furthermore, reading em9c's review raised a new question about the theoretical foundations of the method. The fact that g depends on a specific trajectory x(t) prevents g from being a proper global transition operator. The paper is a bit fuzzy about how the Lie group structure embeds into the proposed framework. So do we have a Lie group per trajectory? Or a global Lie group? I believe the paper would profit from clarifications on that.

---

> ### Author Response · Authors · 2024-11-27
> **Thank you very much for the followup! (Part I)**
>
> Thank you very much for your thoughtful feedback and for reconsidering the evaluation.
>
> **Regarding your question about the Lie group structure in our framework:**
>
> > The fact that g depends on a specific trajectory x(t) prevents g from being a proper global transition operator. The paper is a bit fuzzy about how the Lie group structure embeds into the proposed framework. So do we have a Lie group per trajectory? Or a global Lie group?
>
> We apologize for any confusion in the original manuscript.  We assume there is a **common (global) Lie group $G$** that acts on the observation space $\mathcal{X}$, and that it can realize any transition in the observed time series across all environments through group action.
>
> To formalize this, consider a dataset of trajectories $\lbrace x_i(t) \mid t \in I \rbrace_i$ from some environment $\mathcal{E}$. Then for each $i$, we assume that there exists a set of Lie group elements $\lbrace g_{i, t, \delta} \mid t \in I \rbrace \subset G$ that can realize the transitions  of $x_i$ through the group action, as follows:
>
> $$
> x_i(t+\delta) = g_{i, t, \delta} \cdot x_i(t), \quad \text{so that} \quad x_i(t) = g_{i, 0, t} \cdot x_i(0).
> $$
>
> Here, we emphasize that $g \in G$ acts on the observation space $\mathcal{X}$ via a group action, mapping any state $x$ to $g \cdot x$. We assume that this holds for all trajectories $x_i$ in all environments including $\mathcal{E}$, so the Lie group $G$ is **common to all trajectories and environments** in our framework.
>
> As we notified all reviewers, we also made sure to clarify this point in the revised section 3 and in Appendix.
>
> **Additional remarks:**
>
> We would also like to reiterate our response to Reviewer em9c regarding our use of Lie group action.  While the elements of $G$ are invertible (reversible), we would like to emphasize that our framework **can** model **non-reversible action controls** as well because the group elements $g(t)$ are time-dependent, and because our controller interface takes into account both the history of action signals $a[0, t)$ and states $x[0, t)$ to output an operator $g(t)$.
>
> For example, we can train our controller so that, at a certain state in a certain environment at a given time $t$, no action input can cause the controller to output the inverse $g(t-1)^{-1}$ of the previous transition operator $g(t-1)$. This situation essentially demonstrates an example of an "action input that cannot be undone," and it is fully within the scope of our theoretical framework and the design of our controller.
>
> We note that  our model **without the controller component (CIP)** can be thought of as a "debug room" where any action can be reversed. However, having such reversibility in the latent space does not impede our ability to model irreversible phenomena. As shown in the video included with our submission, we have demonstrated that we can model phenomena involving diverse types of transitions.
>
> **Addressing any additional concerns:**
>
> > They address some (yet not all) of my concerns.
>
> We appreciate your acknowledgment that some of your concerns have been addressed. If there are any remaining issues or questions, we are fully committed to resolving them. We would greatly appreciate it if you could specify any additional points you'd like us to clarify.
>
> Thank you once again for your valuable feedback, which has helped us improve the clarity and quality of our work.

---

> ### Author Response · Authors · 2024-11-28
> **Thank you very much for the followup! (Part II)**
>
> To address your concern about the significance (statistical test / error bar), we conducted additional experiments by re-running our tests with five different random seeds. We calculated the mean and standard deviation for each metric. As you can see from the standard deviations, the variations are quite small, indicating the stability of our results. These experiments were conducted under the same settings as those in Table 2 of our paper.
>
> | Environment | Metric       |     1  |     2  |     3  |     4  |     5  |  Mean  |  Std   |
> |-------------|--------------|-------:|-------:|-------:|-------:|-------:|-------:|-------:|
> | bigfish     | $\Delta_t$PSNR |   1.24 |   1.25 |   1.24 |   1.27 |   1.31 |   1.26 |  0.03 |
> |             | LPIPS        |   0.04 |   0.04 |   0.04 |   0.04 |   0.04 |   0.04 |  0.00 |
> |             | PSNR         |  24.01 |  24.00 |  23.98 |  24.06 |  24.12 |  24.03 |  0.06 |
> | bossfight   | $\Delta_t$PSNR |   0.31 |   0.32 |   0.33 |   0.33 |   0.35 |   0.33 |  0.01 |
> |             | LPIPS        |   0.19 |   0.19 |   0.19 |   0.19 |   0.19 |   0.19 |  0.00 |
> |             | PSNR         |  18.53 |  18.51 |  18.52 |  18.53 |  18.54 |  18.52 |  0.01 |
> | caveflyer   | $\Delta_t$PSNR |   2.60 |   2.63 |   2.65 |   2.65 |   2.68 |   2.64 |  0.03 |
> |             | LPIPS        |   0.17 |   0.17 |   0.17 |   0.17 |   0.17 |   0.17 |  0.00 |
> |             | PSNR         |  17.78 |  17.74 |  17.78 |  17.79 |  17.77 |  17.77 |  0.02 |
> | climber     | $\Delta_t$PSNR |   2.86 |   2.75 |   2.72 |   2.85 |   2.89 |   2.81 |  0.07 |
> |             | LPIPS        |   0.18 |   0.18 |   0.18 |   0.17 |   0.18 |   0.18 |  0.00 |
> |             | PSNR         |  19.65 |  19.54 |  19.57 |  19.64 |  19.65 |  19.61 |  0.05 |
> | coinrun     | $\Delta_t$PSNR |   9.35 |   9.18 |   9.12 |   9.16 |   9.20 |   9.20 |  0.09 |
> |             | LPIPS        |   0.05 |   0.05 |   0.05 |   0.05 |   0.05 |   0.05 |  0.00 |
> |             | PSNR         |  22.33 |  22.30 |  22.26 |  22.29 |  22.32 |  22.30 |  0.03 |
> | maze        | $\Delta_t$PSNR |   1.63 |   1.62 |   1.57 |   1.53 |   1.64 |   1.60 |  0.05 |
> |             | LPIPS        |   0.12 |   0.12 |   0.12 |   0.12 |   0.12 |   0.12 |  0.00 |
> |             | PSNR         |  21.75 |  21.75 |  21.73 |  21.75 |  21.78 |  21.75 |  0.02 |
> | miner       | $\Delta_t$PSNR |   1.32 |   1.31 |   1.30 |   1.31 |   1.36 |   1.32 |  0.02 |
> |             | LPIPS        |   0.09 |   0.09 |   0.09 |   0.09 |   0.09 |   0.09 |  0.00 |
> |             | PSNR         |  21.84 |  21.81 |  21.79 |  21.80 |  21.83 |  21.81 |  0.02 |
> | ninja       | $\Delta_t$PSNR |   4.14 |   4.10 |   4.14 |   4.10 |   4.14 |   4.12 |  0.02 |
> |             | LPIPS        |   0.18 |   0.18 |   0.18 |   0.18 |   0.18 |   0.18 |  0.00 |
> |             | PSNR         |  19.74 |  19.67 |  19.68 |  19.70 |  19.72 |  19.70 |  0.03 |
>
> We hope these additional results address your concern. Please let us know if you have any further questions or suggestions.

---

> > ### Author Response · Authors · 2024-12-02
> >
> > Dear Reviewer N9Uv,
> >
> > Thank you for your thoughtful questions and valuable feedback on our paper. We have provided detailed responses to address your concerns and have made revisions where appropriate. Please let us know if you have any further comments or if there's anything else we can clarify before the discussion phase concludes in two days.

---

### Author Response · Authors · 2024-11-20
**General Response G1. Additional Experiment on the 1X World Model Dataset**

To further validate our method, we compared WLA against Genie on the **1X World Model dataset**[^1]. This dataset comprises over 100 hours of videos capturing the actions of android-type robots in various environments and lighting conditions, including narrow hallways, spacious workbenches, and tabletops with multiple objects. We slightly adapted the architecture of our method to suit this setting, as detailed below, but otherwise followed the same experimental protocol as with the ProcGen dataset.

### **Results**

On the 1X World Model dataset, we observed results consistent with those from the ProcGen dataset. While Genie produces cleaner predictions for individual frames, it fails to generate video sequences that align with the provided action sequences. In contrast, our method successfully learns the correct responses, producing videos that more closely resemble the ground truth.

These qualitative observations are supported by the quantitative results shown in the table below. We evaluated the models using three metrics:

1. **Peak Signal-to-Noise Ratio (PSNR)**
2. **Temporal Difference PSNR ($\Delta_t$PSNR)**
3. **Fréchet Video Distance (FVD)**[^2], utilizing the debiased version[^3]

Although our method slightly underperforms Genie in frame-wise evaluation (PSNR), it achieves better performance in temporally local evaluation ($\Delta_t$PSNR) and significantly outperforms Genie in FVD. This indicates that our generated videos are more natural and closer to the ground truth distribution.

We will include some of the predicted videos in the supplementary material.

#### **Quantitative Results on the 1X World Model Dataset**

|        | PSNR (↑)    | $\Delta_t$PSNR (↑) | FVD (↓)     |
|--------|-------------|---------------------|--------------|
| Genie  | **21.16**   | 0.78                | 393.85       |
| Ours   | 20.82       | **1.13**            | **131.02**   |

### **Conclusion**

These results demonstrate that WLA is capable of modeling real-world robot actions in 3D environments, in addition to 2D game environments. This enhancement addresses the concerns about limited experimental evaluation and increases trust in our method and framework.

### **Implementation Details**

In this experiment, each $256 \times 256$ RGB video frame was converted into an $8 \times 8$ grid of 18-bit binary tokens using MAGVIT2. Consequently, we trained our model on the time series of these tokens and replaced the reconstruction losses with logistic losses, as the tokens are binary. Additionally, since Slot Attention expects images as input, we employed a standard Vision Transformer (ViT) architecture to model the time evolution and alignment of the encoded tokens. Specifically, since there are $8 \times 8 = 64$ tokens in total, we treat them as if they are “slot” tokens. Also, we set the number of rotation angles to $J = 16$.

---

[^1]: Available at [https://huggingface.co/datasets/1x-technologies/worldmodel](https://huggingface.co/datasets/1x-technologies/worldmodel); we used version 1.1.

[^2]: Unterthiner et al., "Towards Accurate Generative Models of Video: A New Metric & Challenges," 2024.

[^3]: Ge et al., "Content-Debiased FVD for Evaluating Video Generation Models," 2024.

---

### Author Response · Authors · 2024-11-20
**General Response G2. Hyperparameter Selection**

Reviewers N9Uv and vJwu raised questions and concerns on the choice of hyperparameters, specifically the number of slots $N$ and the number of Lie group actions $J$. We agree that these are crucial and will include a detailed discussion in both the main text and the Appendix.

Generally, increasing $N$ and $J$ enhances the model capacity but also affects computational complexity, especially $N$, which directly impacts runtime. The same applies to $J$ since $D = 2J$.

We observed that the internal construction of $D$ significantly improves performance. Inspired by sinusoidal positional embeddings and G-NFT, we parameterized each rotation with a "frequency." We indexed $\theta$ with $i, j$ such that $\theta_{i, j}(t) = m_i \tilde{\theta}_j(t)$, where $\lbrace m_i \in \mathbb{R} \mid i = 1, \dots, F \rbrace$ are the rotation speeds and $j \in 1, \dots, \tilde{J}$ with $\tilde{J} = J / F$. We fixed $F$ with $D = 2J = \tilde{J} F$.

We conducted additional small-scale experiments on ProcGen to study the effect of varying $\tilde{J}$ and $N$. The results are summarized in the following table:

| $N$  | $\tilde{J}$ | Reconstruction Error (MSE) |
|------|-------------|----------------------------|
| 16   | 3           | 0.0583                     |
| 16   | 6           | 0.0587                     |
| 16   | 12          | 0.0439                     |
| 16   | 24          | 0.0378                     |
| 8    | 6           | 0.0641                     |
| 16   | 6           | 0.0587                     |
| 24   | 6           | 0.0397                     |

These results indicate that increasing $\tilde{J}$ with a smaller $F$ improves performance. Similarly, increasing $N$ reduces reconstruction error, highlighting the importance of these hyperparameters.

---

### Author Response · Authors · 2024-11-26
**Revised Paper Submitted**

We have revised our paper based on your valuable comments. The main changes are as follows:

- **Added new experiments** (Section 6.3)
- **Improved figures** illustrating the model overview (Figures 1 and 2)
- **Enhanced mathematical rigor** (Sections 2, 3, 4, and Appendix A)
- **Included additional related work** (Section 5)
- **Added discussion on hyperparameters** (Section 4.4 and Appendix B4)
- **Corrected typos**

The major revisions are highlighted in blue in the manuscript.

We have also included videos of the new experiments in the supplementary materials. These videos clearly demonstrate the differences compared to the baseline methods, and we hope they provide additional support to the significance of our framework.

---

### Meta-Review · Area_Chair_KE8n · 2024-12-20

**Metareview:**

This paper introduces World modeling through Lie Action (WLA), a method for learning action-conditioned video prediction models using an object-centric latent representation. It learns a continuous latent action space that generalises across environments.

Overall, the reviewers positively highlighted the clarity of the paper and the novelty of the theoretically-inspired framework.

There were some concerns about the validity of the theoretical assumptions made in the paper, but these were well addressed during the rebuttal. The main concerns raised by the reviewers, however, related to the quality and breadth of the experimental evaluation (simplistic tasks [PHYRE, ProcGen], single baseline).

The newly added robotics video experiments are a step in the right direction, but the evaluation setting considered is too short (in terms of predicted time horizon) to be very meaningful. For example, the video results in the appendix capture primarily static scenes with little to no motion over a very short time horizon that should be very easy to model. Given that these results were only added during the rebuttal period, it would be good to further refine these and get them properly reviewed at another conference.

To have a better chance of being considered at a top-tier conference like ICLR, the authors could, for example, also spend more effort on validating the method against an existing model on an existing (challenging) task to properly ground the performance of their method.

**Additional Comments On Reviewer Discussion:**

Discussion did not affect reviewer opinion. No reviewer was willing to champion the paper for acceptance.

---

### Decision · Program_Chairs · 2025-01-22

Reject